# Invariance to background noise as a signature of non-primary auditory cortex

Alexander J.E. Kell[1,2,3,5] & Josh H. McDermott [1,2,3,4]

Despite well-established anatomical differences between primary and non-primary auditory cortex, the associated representational transformations have remained elusive. Here we show that primary and non-primary auditory cortex are differentiated by their invariance to real-world background noise. We measured fMRI responses to natural sounds presented in isolation and in real-world noise, quantifying invariance as the correlation between the two responses for individual voxels. Non-primary areas were substantially more noise-invariant than primary areas. This primary-nonprimary difference occurred both for speech and non-speech sounds and was unaffected by a concurrent demanding visual task, suggesting that the observed invariance is not specific to speech processing and is robust to inattention. The difference was most pronounced for real-world background noise—both primary and non-primary areas were relatively robust to simple types of synthetic noise. Our results suggest a general representational transformation between auditory cortical stages, illustrating a representational consequence of hierarchical organization in the auditory system.

[1] Department of Brain and Cognitive Sciences, MIT, Cambridge, MA 02139, USA. [2] McGovern Institute for Brain Research, MIT, Cambridge, MA 02139, USA. [3] Center for Brains, Minds, and Machines, MIT, Cambridge, MA 02139, USA. [4] Program in Speech and Hearing Biosciences and Technology, Harvard University, Boston, MA, USA. [5] Present address: Zuckerman Institute of Mind, Brain, and Behavior, Columbia University, New York, NY 10027, USA. Correspondence and requests for materials should be addressed to A.J.E.K. (email: alexkell@mit.edu) or to J.H.M. (email: jhm@mit.edu)

Humans and other animals infer a rich array of information about the world from sound. Much of this information is not readily accessible from the sound pressure waveforms impinging on the eardrums. A cascade of sensory processing is thought to be required to transform the incoming waveforms into a representational format where behaviorally relevant information is made explicit[1–3]. This cascade begins with the cochlea, continues through the auditory midbrain and thalamus, and culminates in what are believed to be multiple stages of cortical processing[3–9]. Within the auditory cortex, primary and non-primary areas exhibit clear anatomical differences in cytoarchitecture and connectivity[10,11]. Physiologically, non-primary neurons tend, on average, to exhibit longer response latencies[4,5,12] and broader frequency tuning[4,13,14]. However, the consequences of such differences for real-world hearing—i.e., the concomitant representational transformations—remain unclear.

Previous attempts to characterize such transformations have largely been limited to speech[2,15–21], which may involve a specialized pathway[22–24], or have suggested that non-primary areas are more influenced by task or attention[4,17,25]. Here, we probe for a general sensory transformation that might differentiate stages of representation, measuring the invariance of sound-evoked responses throughout the auditory cortex to the presence of background noise.

Listening in noise is a core problem in everyday hearing. Given that sounds rarely occur in isolation and sum linearly in the air, sounds of interest are often superimposed on a background of noise from other sound sources. When, for instance, you are in a bustling coffee shop and hear your phone ring or listen to a companion speak, the ambient background noise distorts the pattern of spikes in the auditory nerve, often substantially. Thus, to recognize sources of interest, the auditory system must somehow separate or suppress the effects of the noise. Typical human hearing is remarkably noise-robust, but listeners with age-related hearing loss or other forms of impaired hearing struggle in noisy environments—and are not much helped by contemporary hearing aids[26,27].

What is considered "noise" can depend on the context, and thus the ability to hear sound sources of interest is in some cases critically dependent on selective attention[28–32]. However, some sounds are more informative than others on basic statistical grounds. Stationary signals, for instance, have stable statistics over time and thus convey little new information about the world, such that it might be adaptive to attenuate their representation relative to nonstationary sounds. There has been long-standing interest within engineering in developing methods to separate or remove these less informative (stationary) sounds from audio signals[33,34]. Neuroscientists have explored the possibility that the brain might by default do something similar, measuring how neural responses are affected by simple synthetic noise[15,18,19,35–39]. Here, we seek to extend the logic implicit in this research tradition to real-world sounds[40,41], using a measure of stationarity to characterize the extent to which natural sounds are noise-like in this statistical sense. Stationary sounds, as are produced by rain, fire, swarms of insects etc., are commonly referred to as "textures"[41]. Although the perception of isolated textures has been characterized to some extent[41–43], little is known about their representation in the brain, or about how they affect the perception of concurrent foreground sounds[43]. We hypothesized that robustness to these more structured sources of everyday noise might necessitate mechanisms situated later in the putative cortical hierarchy.

We assess the invariance of cortical responses to a broad sample of everyday sounds embedded in various sources of real-world noise, taking advantage of the large-scale coverage afforded by fMRI to examine this invariance across multiple cortical fields throughout the human auditory cortex. We find that responses in primary areas are substantially altered by real-world background noise, whereas responses in non-primary areas are more noise invariant. We show that this pattern of invariance is general (i.e., it is not specific to speech or music processing), that it is not observed for simpler synthetic background noises that lack the structure of many real-world sounds, and that it is robust to inattention. The results suggest mechanisms of noise robustness that are complementary to those of directed attention, boosting responses to nonstationary signals in auditory scenes. More broadly, these findings reveal a general transformation between auditory cortical stages, illustrating a representational consequence of the putative hierarchical organization in the auditory system that has clear relevance to everyday perception and behavior.

## Results

**Measuring the stationarity of real-world sounds.** To select real-world background "noise", we measured the stability of sound properties (i.e., the stationarity) of hundreds of natural sounds, assigning sounds with the most stationary statistics to a background noise set (Fig. 1a). We first computed a cochleagram of each sound—a time–frequency decomposition obtained from the output of a simulated cochlear filter bank. We divided the cochleagram into small temporal segments, and in each segment separately measured three sets of perceptually relevant statistics[41–43]: (i) the mean of each frequency channel (capturing the spectrum), (ii) the correlation across frequency channels (capturing aspects of co-modulation), and (iii) the power in a set of temporal modulation filters applied to each frequency channel (capturing rates of amplitude modulation). To assess stationarity at different time scales, we varied the segment length (50, 100, and 200 ms). We computed the standard deviation of each of these statistics over time, averaging across statistics and segment lengths to return a single stationarity score for each sound (see the Methods section for details). We used sounds with the most stationary statistics as noise sources for the experiments below (the resulting sounds are those that would typically be considered textures; Fig. 1b lists the stimuli for Experiment 1; see Supplementary Fig. S1 for summary acoustic measures of the stimuli).

**Noise invariance increases from primary to non-primary areas.** We examined the invariance of auditory cortical representations to background noise by assessing the extent to which noise altered cortical responses to natural sounds. In Experiment 1, we measured fMRI responses in 11 human listeners to 30 natural sounds presented in quiet, as well as embedded in 30 real-world background noises (Fig. 1c). To minimize the MR image acquisition time, we used a partial field of view, recording responses throughout the temporal lobe (Fig. 1d). Each sound was ~2 s long (see Supplementary Fig. S2 for schematic of the experimental design). To quantify the invariance of the cortical response, we leveraged the fact that a voxel's response typically varies across natural sounds, presumably due to the tuning properties of the neurons sampled by the voxel. We simply correlated each voxel's response to the natural sounds in isolation with its response to those same natural sounds embedded in background noise (Fig. 1c). If the neurons sampled by a voxel are robust to background noise, this correlation coefficient should be high. Different voxels exhibit different amounts of measurement noise (Supplementary Fig. S3), due to a variety of factors, including differences in distance from the receiver coils. To enable comparisons across voxels, we corrected for such measurement noise[44,45] (see the Methods section for details). As a result of this reliability correction, any differences in the robustness metric across cortical

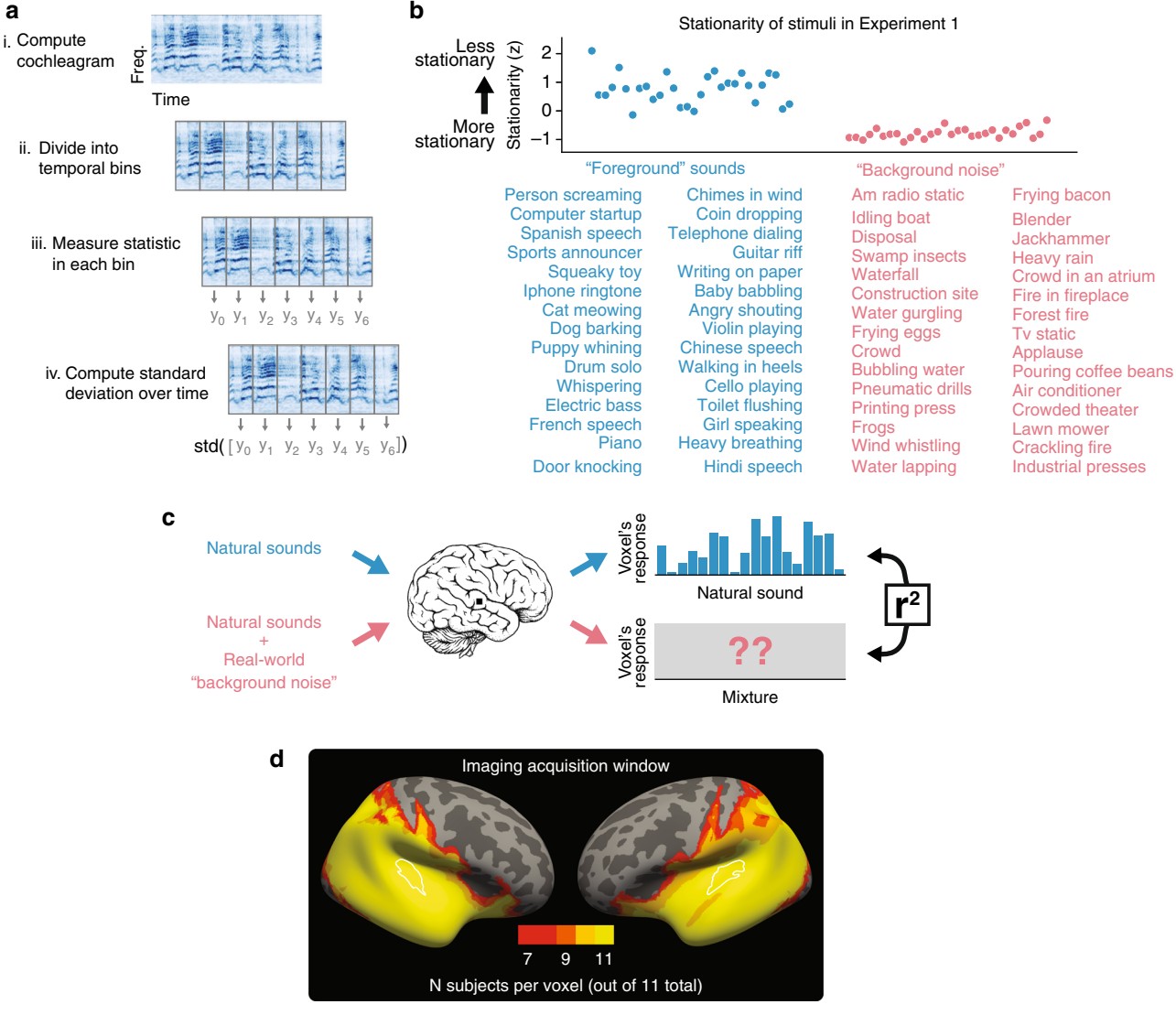

**Fig. 1** Experimental design. **a** Schematic of stationarity score used to select real-world "background noise." **b** Sounds selected by stationarity. Top: stationarity score for each sound in Experiment 1. Tick marks indicate z-score units. Bottom: list of all 30 natural sounds, and all 30 real-world background noises used in Experiment 1. **c** Schematic of stimulus logic and analysis. **d** Heatmap of field of view in Experiment 1

areas is not the result of differences in measurement noise across cortical areas.

Primary cortical responses were substantially altered by background noise, but non-primary responses were more robust. Figure 2a shows summary maps of the invariance metric in all voxels that exhibited a reliable response to sounds, averaged across all 11 participants. A two-tailed paired $t$ test showed a significant difference in mean invariance between a primary and non-primary region of interest (dark blue line in Fig. 2b; $t_{10} = 8.37$, $p = 7.88 \times 10^{-6}$; see Supplementary Fig. S4 for plots with individual subjects). The difference in the effect of background noise on the invariance of responses in the primary and non-primary regions could not be explained simply by differential suppression of responses by background noise—overall responses were similar in the two conditions (Supplementary Fig. S5). These results suggest that invariance to real-world background noise may distinguish primary and non-primary auditory cortical responses.

**Increased invariance is not specific to speech or music**. It has long been proposed that parts of non-primary human auditory cortex may serve to produce invariant representations of

speech[15,17–19]. But is speech processing somehow special in this regard? Or is non-primary processing more noise-invariant irrespective of the sound source? Experiment 1 left this question open because about half of the foreground sounds were instances of speech or music. Given the relatively small number of non-speech and non-music sounds in the data from Experiment 1, we ran a second experiment with a new stimulus set.

In Experiment 2, we scanned another 12 participants in a similar paradigm, but with speech and music excluded (Supplementary Table S1 lists the 35 foreground and 35 background sounds used). The pattern of invariance was similar to that observed in Experiment 1 (Fig. 2c). A two-tailed paired $t$ test again showed a difference in the mean invariance in primary and non-primary regions of interest (light blue line in Fig. 2b; $t_{11} = 4.56$, $p = 8.17 \times 10^{-4}$). An ANOVA comparing Experiment 1 and 2 showed a main effect of region ($F_{1,26} = 57.4$, $p = 4.83 \times 10^{-8}$), but failed to show a main effect of experiment ($F_{1,26} = 0.02$, $p = 0.88$) or an experiment-by-ROI interaction ($F_{1,26} = 1.39$, $p = 0.25$). These results suggest that the noise invariance of non-primary auditory cortical responses is not simply a reflection of speech and music processing, and may instead be a more generic property of non-primary representations.

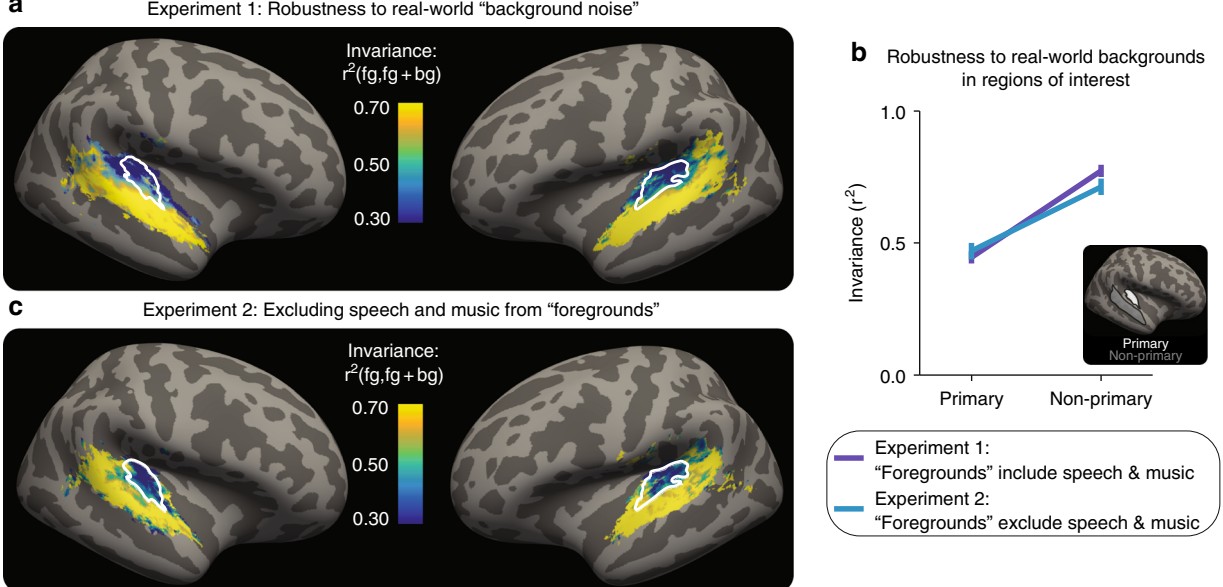

**Fig. 2** Non-primary auditory cortex is robust to real-world background noise. **a** The results of Experiment 1 ($n = 11$): map of invariance metric, averaged across participants. White outlines: primary-like areas, defined anatomically. **b** Median of invariance metric in a primary and non-primary region of interest (ROI), averaged across participants and hemispheres. Inset shows ROIs on the inflated brain. Error bars are within-participants SEM. Inset: outlines of anatomical ROIs. **c** The results of Experiment 2 ($n = 12$). Experiment 2 was similar to Experiment 1, but without speech or music stimuli. Plotting conventions, including colorscale, are identical to (**a**)

**Increased invariance is specific to real-world noise**. How do our results relate to previous work, which used simple, synthetic noise signals[15,18,19,35–37,39]? Given that such synthetic signals lack the structure present in many real-world sounds[40,41], it seemed plausible that invariance to simpler synthetic noise could arise earlier in a putative hierarchical processing stream compared with the invariance to more naturalistic noise sources. To address this question, in Experiment 3, we scanned another 23 participants with the same 35 foreground sounds as Experiment 2 (i.e., without speech or music), but in two different background-noise conditions. In the first condition, we sought to replicate Experiment 2, presenting the same 35 real-world background noises. In the second condition, we replaced each of the 35 real-world background noises with a spectrally matched Gaussian noise signal—i.e., with the same long-term power spectrum as the corresponding real-world noise, but otherwise unstructured.

Figure 3a shows example cochleagrams of the real-world and spectrally matched synthetic backgrounds (see Supplementary Fig. S1 for the average modulation spectrum of these sounds). Although these synthetic noise signals match the spectrum of the real-world background noise, they lack the higher-order statistical structure present in the real-world noises. Figure 3b plots two example of statistic classes that characterize natural sound textures[41–43]: the cross-channel correlation matrix (left panels) and modulation spectrum (right panels). Both classes of statistic vary across real-world noises, but not the synthetic noises, for which they instead resemble the statistics of white noise (which provides a completely unstructured point of reference, Fig. 3b, far right; see Supplementary Fig. S6 for quantification of these differences).

The results with the real-world background noise replicated Experiment 2 (Experiment 3, Condition 1; Fig. 3c): non-primary areas were substantially more robust to real-world background noise than primary areas (light blue line in Fig. 3d; two-tailed paired $t$ test: $t_{22} = 7.40$, $p = 2.10 \times 10^{-7}$; see Supplementary Fig. S7 for analyses of subregions of primary auditory cortex). By contrast, when those same natural sounds were embedded in

synthetic noise, responses in both primary and non-primary areas were relatively invariant (Experiment 3, Condition 2; Fig. 3e and red line in Fig. 3d): a two-way ANOVA showed an interaction of noise type (real-world versus synthetic) by ROI (primary versus non-primary) ($F_{1,66} = 5.39$, $p = 0.023$). We obtained qualitatively similar results when we examined the robustness of patterns of responses across voxels to the presence of the two types of background noise; multi-voxel patterns in non-primary areas exhibited greater invariance than those in primary areas (Supplementary Fig. S8). Taken together, these results are consistent with evidence in nonhuman animals of invariance to synthetic noise in primary auditory cortex[35,37,39] and demonstrate that primary and non-primary areas may be specifically distinguished by their invariance to more realistic sources of background noise.

**Cortical invariance is relatively unaffected by inattention**. "Foreground" sounds were selected for their nonstationarity, and it therefore seemed plausible that they could be more salient and thus preferentially draw participants' attention[46,47]. Because attentional modulations may be stronger in non-primary compared with primary areas[4,17,25], it seemed possible that non-primary areas could appear more invariant than primary areas simply because they are more strongly altered by attention. We examined this possibility in Experiment 4, measuring the invariance of auditory cortical responses while attention was directed to an auditory task or a visual task. We chose to manipulate attention across modalities because we had difficulty devising a task that could verifiably direct attention within the auditory modality to the foreground sound or background noise.

During the auditory task, participants ($n = 21$) performed the same task as in the previous three experiments: detecting differences in the intensity of successive sound stimuli. During the visual task, participants performed a demanding one-back task (Fig. 4a), on a stream of four-by-four grids (each displayed for ~700 ms with an interstimulus interval of ~100 ms). In each

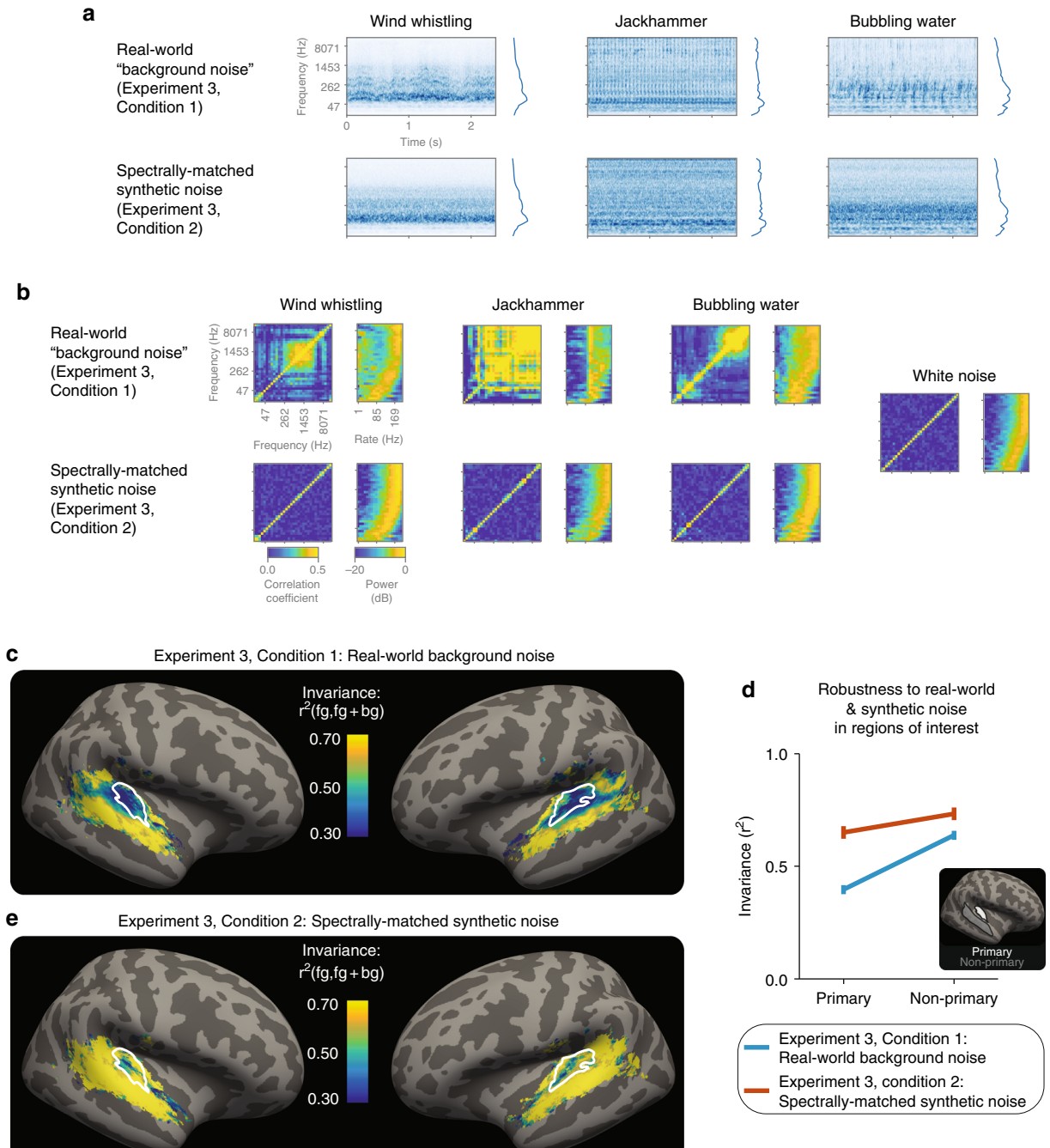

**Fig. 3** Non-primary areas are particularly robust to real-world background noise. **a** Example cochleagrams of real-world background noises and spectrally matched synthetic noises. Average spectrum of each sound is shown to the right of each cochleagram. **b** Statistics of example real-world noises (top row) and spectrally matched synthetic noises (bottom row). Left subpanel show cross-channel envelope correlations; right subpanel shows modulation spectra, measured as power in a bank of modulation filters applied to each cochlear envelope. **c** The results of Experiment 3, Condition 1 ($n = 23$): map of invariance metric for real-world background noises, averaged across participants. White outlines: primary-like areas, defined anatomically. **d** Median of invariance metric in a primary and non-primary region of interest (ROI), averaged across participants and hemispheres. Error bars are within-participants SEM. Inset: outlines of anatomical ROIs. **e** The results of Experiment 3, Condition 2 ($n = 23$): map of invariance metric for spectrally matched synthetic noises, averaged across participants. Plotting conventions, including colorscale, are identical to (**c**)

grid, six of the sixteen squares were colored; the location of two of the six colored squares changed from stimulus to stimulus unless the pattern repeated. Participants had to detect exact repeats, and report these repeats with a button press within the ~700 ms display period. Participants performed both tasks well (d-prime on visual task: mean of 3.3, range of 1.8–4.5; accuracy on auditory task: mean of 0.92, range of 0.83–0.99). The stimuli presented during each task condition were identical, as the stream of grids was displayed both during both tasks. The only difference between the two conditions was which task the participants were performing.

As intended and expected[48,49], the tasks modulated overall response magnitudes in auditory and visual cortex. Mean responses in the auditory cortex were higher during the auditory

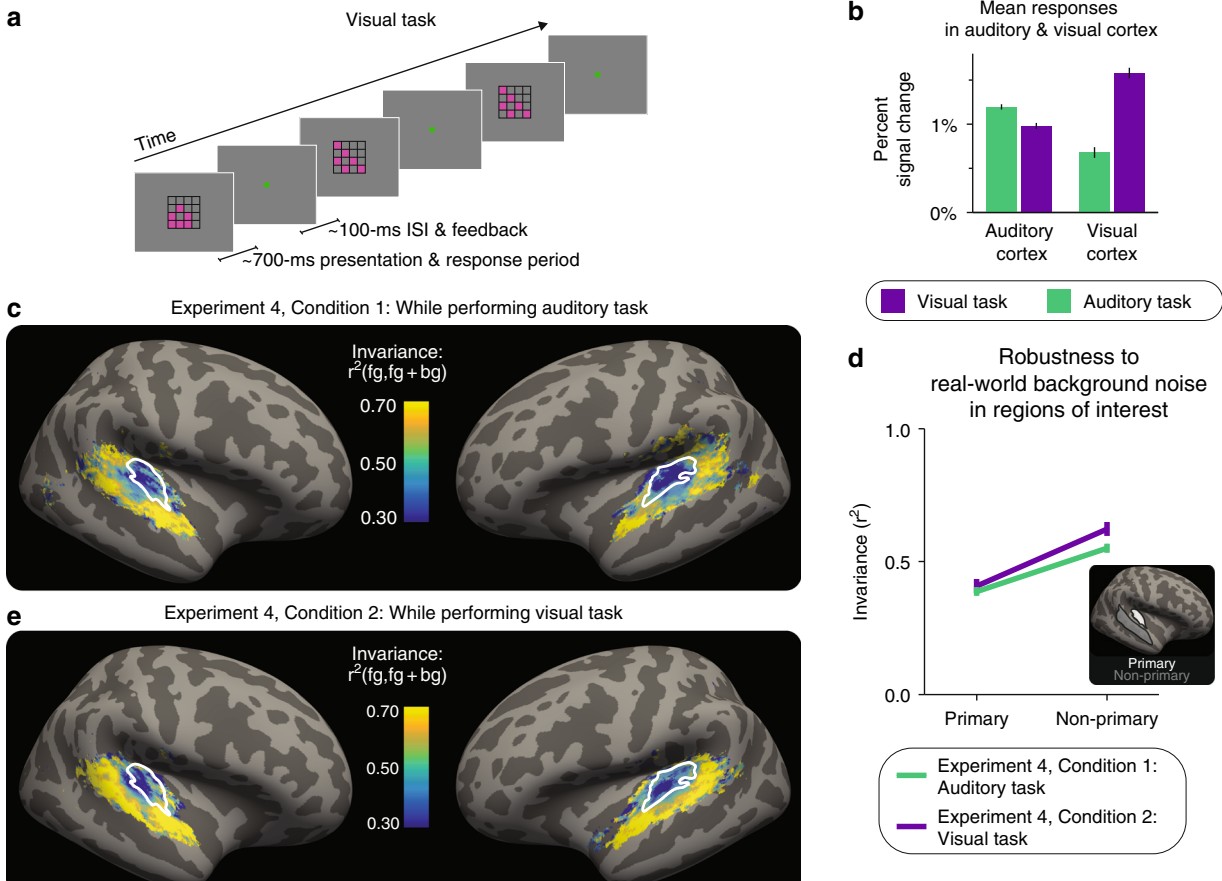

**Fig. 4** Increased invariance in non-primary areas is robust to inattention. **a** Schematic of visual task. **b** Mean responses in auditory and visual cortex during each task. Error bars are within-participant SEM. **c** The results of Experiment 4, Condition 1 ($n = 21$): map of invariance metric for real-world background noises during the auditory task, averaged across participants and hemispheres. White outlines: primary-like areas, defined anatomically. **d** Median of invariance metric in a primary and non-primary region of interest (ROI), averaged across participants and hemispheres. Error bars are within-participants SEM. Inset: outlines of anatomical ROIs. **e** The results of Experiment 4, Condition 2 ($n = 21$): map of invariance metric for real-world background noises during the visual task, averaged across participants. Plotting conventions, including colorscale, are identical to (**c**)

task than the visual task, and mean responses in the visual cortex were higher during the visual task than the auditory task (Fig. 4b; two-tailed paired $t$ tests, respectively, $t_{20} = 5.76$, $p = 1.23 \times 10^{-5}$ and $t_{20} = 9.78$, $p = 4.58 \times 10^{-9}$; ANOVA: region-by-task interaction: $F_{1,60} = 142$, $p = 1.11 \times 10^{-16}$). Mean responses in the visual cortex were more substantially altered by task, perhaps because eye movements may have systematically differed across tasks, as participants were not required to fixate in either condition. Within auditory cortex, there was a trend for mean responses in non-primary areas to be more affected by the attentional task than those in primary areas, but the interaction did not reach significance (Supplementary Fig. S9; ANOVA: $F_{1,60} = 3.34$, $p = 0.0728$). Overall, these results indicate that the tasks achieved the objective of manipulating attention, as indexed by the differential mean responses across tasks.

We then compared the cortical invariance during the two tasks. During the auditory task, non-primary areas were again substantially more robust than primary areas (Fig. 4c; a two-tailed paired $t$ test comparing mean invariance across ROIs: $t_{20} = 5.87$, $p = 9.53 \times 10^{-6}$; green line in Fig. 4d). However, a similar pattern of robustness was observed during the visual task (Fig. 4e; a two-tailed paired $t$ test comparing mean invariance across ROIs: $t_{20} = 4.74$, $p = 1.25 \times 10^{-4}$; purple line in Fig. 4d). Moreover, the auditory task did not significantly alter the robustness in primary or non-primary areas: an ANOVA revealed a main effect of region (primary/non-primary: $F_{1,60} = 48.3$, $p = 3.05 \times 10^{-9}$), but

neither the task main effect nor a task-by-region interaction was significant (respectively: $F_{1,60} = 1.64$, $p = 0.21$ and $F_{1,60} = 1.59$, $p = 0.21$). Taken together, the results from Experiment 4 suggest that while attention alters the mean responses in the auditory cortex, it does not account for the increase in invariance between primary and non-primary auditory cortex. The increased invariance of non-primary auditory cortex to background noises thus appears to be robust to inattention.

## Discussion

Taking advantage of the large-scale coverage afforded by fMRI, we found that non-primary auditory cortex was substantially more invariant to real-world background noise than primary auditory cortex. This increase in invariance was not specific to speech or music processing—it occurred for other real-world sounds as well, suggesting that it may be a generic property of non-primary representations of sound. This primary/non-primary difference was weaker for simpler, synthetic noise that lacked the structure typical of many natural sounds. Lastly, this difference in invariance was robust to inattention, suggesting that attention does not account for the effect. These results demonstrate that noise invariance may be a generic functional signature of non-primary auditory cortex, illustrating a representational consequence of the putative hierarchical organization in auditory cortex that has relevance to perception and to behavior.

Listening in noise is a core problem of real-world hearing, and there has been long-standing interest in how auditory cortex robustly encodes sounds in the presence of background noise[15,17–19,21,35–37,39,50]. Our work makes three contributions.

First, we demonstrate robustness to real-world sources of stationary noise, and in the process reveal differentiation between regions of the auditory cortex. Previous work was primarily restricted to simpler, synthetic noise[15,18,19,35–37,39], which lack the structure of many natural background noises[40,41]. We found that primary areas were relatively invariant to such simpler, synthetic noise, consistent with this previous work. However, we also found that primary responses were not particularly invariant to more realistic sources of background noise (textures). Invariance to those more realistic sources of noise arose only in non-primary auditory cortex, later in the putative cortical hierarchy.

Second, prior work on problems of auditory invariance more generally has largely been focused on speech. Although there are many prior reports that parts of non-primary auditory cortex are invariant to sources of acoustic variability in speech[2,15,16,18,19,21], little has been known about the invariance of non-primary cortical responses to non-speech sounds. We show that non-primary processing of a wide variety of real-world sounds is robust to background noise.

Third, we show that noise robustness is present even when attention is directed elsewhere. Non-primary cortical responses have been observed to be modulated by attention to one speech stream over another[17,29], illustrating one way in which cortical responses can achieve invariance to behaviorally irrelevant sounds. It has remained unclear whether robustness to temporally stationary noise might be explained in this way, particularly given that nonstationary sounds might preferentially draw selective attention[46,47]. The effects we report here are relatively unaffected by inattention and thus suggest a form of noise robustness largely distinct from the attentional selection of task-relevant signals. The invariance observed here may be more akin to previously reported aspects of sound segregation that are also robust to inattention[51,52] and/or to engineering algorithms that remove stationary noise from signals[33,34]. However, because we manipulated attention across modalities, rather than within audition, we cannot exclude the possibility that residual attentional resources are captured more by nonstationary sounds, and that this may contribute to the effects we observed.

The results suggest several lines of future work. Most notably, the mechanisms underlying noise robustness are natural targets for computational models[3,35,47,53]. Because the observed noise invariance was present during inattention, it seems unlikely that attentional mechanisms underlie the robustness we study here. Alternatively, could noise invariance be explained by tuning to different rates and scales of spectrotemporal modulation? Because modulation tuning is prominent in primary auditory cortex[54–58], and because stationary and nonstationary sounds have distinct modulation spectra (Supplementary Fig. S1), invariance to noise could result from such noise falling "out of band"[36] with respect to this tuning. Consistent with this general idea, mean responses were overall higher to foreground sounds than background sounds, and the ratio between responses was higher in non-primary auditory cortex (Supplementary Fig. S10). Modeling could clarify whether the difference in invariance for synthetic and more realistic noise sources could be explained in this way. However, given that modulation tuning accounts for little of the neural responses outside primary auditory cortex in humans[57], and given that modulation spectra of real-world sounds vary considerably from sound to sound[41] (Fig. 3b), it seems unlikely that the increase in noise invariance between primary and non-primary cortex could be entirely explained with modulation filtering. Instead, one appealing candidate mechanism for non-

primary noise invariance is some form of adaptation[21,35,47,59,60] that could attenuate the representation of stationary sound components, generating noise-robust representations.

It could also be revealing to use our methods to assess noise robustness in atypical listeners, who exhibit difficulties with listening in noise. For example, examining whether individuals with auditory processing disorder exhibit typical patterns of cortical robustness may help clarify whether the disorder is sensory in nature or whether it is instead a consequence of abnormal executive function[61,62]. Moreover, given the increasing interest in auditory deficits that are not revealed in a conventional audiogram[63,64] (so-called "hidden" hearing loss), it could be fruitful to examine whether such deficits also manifest themselves in diminished cortical noise robustness. Lastly, the experimental design we employ here—measuring the similarity between brain responses to the same source signals in different conditions—could be used to study other kinds of invariances in the auditory system (e.g., intensity[65], reverberation[66], etc.) and in other sensory systems, where similar invariant recognition challenges arise.

## Methods

**Participants**. Each participant for each experiment completed both: (1) a 2 h scan session and (2) a separate behavioral session, in which an audiogram was measured to ensure normal hearing (we required mean pure tone detection thresholds below 15 dB HL for each ear across all frequencies tested: 125, 250, 500, 750, 1000, 1500, 2000, 3000, 4000, 6000, and 8000 Hz). All participants provided informed consent, the work complied with all relevant ethical regulations, and the Massachusetts Institute of Technology Committee on the Use of Humans as Experimental Subjects (COUHES) approved all experiments. Participant demographics were as follows:

Experiment 1: 11 participants (3 female); mean age: 25.5 years (range 20–30)
Experiment 2: 12 participants (8 female); mean age: 27.2 years (range: 22–32)
Experiment 3: 23 participants (12 female); mean age: 24.7 years (range: 19–36)
Experiment 4: 21 participants (13 female); mean age: 25.6 years (range: 19–37).

Some participants partook in multiple experiments, yielding a total of 45 unique participants: 30 participated only in one experiment, nine participated in two experiments (one of which was author A.K.), five participated in three experiments, and one participated in all four.

One additional participant took part in Experiment 1, but was excluded because of high audiometric thresholds for one ear (35 dB HL). Four additional participants took part in Experiment 3, but were excluded: three did not complete the scanning session; the fourth completed the scanning session but had poor behavioral performance (mean performance on two alternative-forced choice in-scanner task: 48%; for other participants, mean performance: 90%, range: 85–97%). Five additional participants took part in Experiment 4: one was excluded because of their audiogram (> 15 dB HL for each ear), two others were excluded because they did not complete the scanning session, and two were excluded for poor behavioral performance (less than 80% correct on the auditory task; for other participants, mean performance: 92%, range: 85–99%).

**Stationarity measures: selection of real-world background noise**. Sound properties were measured from a cochleagram, a time–frequency decomposition of a sound that mimics aspects of cochlear processing. A cochleagram is similar to a spectrogram, but with frequency resolution modeled after the human cochlea, and with a compressive nonlinearity applied to the amplitude in each time–frequency bin. Each sound pressure waveform was passed through a bank of 211 filters (203 bandpass filters along with four highpass and four lowpass to allow for perfect reconstruction). Filters were zero phase with a frequency response equal to the positive portion of a single period of a cosine function. The center frequencies ranged from 30 Hz to 7860 Hz. The filters were evenly spaced on an Equivalent Rectangular Bandwidth (ERB)$_N$ scale, approximately replicating the bandwidths and frequency dependence believed to characterize human cochlear filters[67]. Adjacent filters overlapped in frequency by 87.5%. The envelope of the subband from each filter was computed as the magnitude of the analytic signal (via the Hilbert transform). To simulate basilar membrane compression, these envelopes were raised to the power of 0.3. Compressed envelopes were downsampled to 400 Hz.

We divided the cochleagram into nonoverlapping adjacent time segments, and measured the following sets of statistics in each segment: (i) the mean of each frequency channel (capturing the spectrum); (ii) the correlation across different frequency channels (capturing co-modulation); and (iii) the power in a set of temporal modulation filters applied to the envelopes (capturing rates of amplitude modulation). The filters used to measure modulation power also had half-cosine frequency responses, with frequencies equally space on a log scale (twenty filters, spanning 0.5 to 200 Hz) and a quality factor of 2 (for 3 dB bandwidths). These properties are broadly consistent with previous models of human modulation

filtering[68] and with neurophysiology data from nonhuman animals[69]. For the stationarity measure described below, we only included power from modulation filters whose center frequency completed at least one cycle in each segment (e.g., for the 100-millisecond segments we excluded all filters with center frequencies less than 10 Hz).

For each temporal segment of each cochleagram, we therefore had three variables of interest:

1.  $\mu$, a 211-length vector of the mean of the envelope in each frequency channel
2.  $\Sigma$, a (211,211) matrix of the correlation of the envelopes across frequency channels
3.  $\Gamma$, a (211,20) matrix of the modulation powers at twenty rates for each cochlear channel

We computed each of these variables of interest in each temporal segment. We then took the standard deviation of these variables across temporal segments, resulting in the following:

1.  $\sigma_\mu$, a 211-length vector of the standard deviation across temporal segments of the mean in each frequency channel
2.  $\sigma_\Sigma$, a (211,211) matrix of the standard deviation across temporal segments of correlation of the envelopes across pairs of frequency channels
3.  $\sigma_\Gamma$, a (211,20) matrix of the standard deviation across temporal segments of the modulation power of each frequency channel at each modulation rate

To summarize the temporal stationarity of these three classes of statistics (envelope means, envelope correlations, and envelope modulation power), we simply took the mean of each of these standard deviation variables ($\sigma_\mu$, $\sigma_\Sigma$, and $\sigma_\Gamma$) which resulted in three scalar quantities: $\lambda_\mu$, $\lambda_\Sigma$, and $\lambda_\Gamma$.

We computed each of these quantities ($\lambda_\mu$, $\lambda_\Sigma$, and $\lambda_\Gamma$) for each of 447 sound clips in a diverse library of natural sounds. We z-scored each statistic separately across the 447 clips: $\lambda^z_\mu$, $\lambda^z_\Sigma$, and $\lambda^z_\Gamma$, to put them on a common scale.

To capture the stationarity at different temporal scales, we computed these three quantities ($\lambda^z_\mu$, $\lambda^z_\Sigma$, and $\lambda^z_\Gamma$) using temporal segments of three different lengths: 50, 100, and 200 ms. To yield a single summary measure of a sound's stationarity across different statistics and temporal scales (a stationarity score, $\delta$), we averaged these nine quantities for each sounds ($\lambda^z_\mu$, $\lambda^z_\Sigma$, and $\lambda^z_\Gamma$, computed from the three different segment lengths).

We selected the most and least stationary sound clips under this stationarity score, $\delta$. We excluded some clips in order to increase the diversity of the foreground and background sets (e.g., to avoid too many instances of fire or water in the background set). We also excluded clips if they had long silent periods at the beginning or end—the stationarity metric we employ scores such clips as highly nonstationary irrespective of the stationarity of the sound source.

**Experimental stimuli: foreground sounds and background noise**. The 11 participants in Experiment 1 heard 90 sounds. All 11 heard 30 foreground sounds in isolation, and 30 mixtures of the foreground sounds with real-world background noise. The first four participants additionally heard 30 mixtures of foreground sounds and spectrally matched noise, while the last 7 subjects additionally heard the 30 real-world background noise in isolation. Upon recognizing that more sounds could be fit into a 2-h scan session, we expanded the stimulus sets of Experiments 2 and 3 to include 105 sounds. In Experiment 2, each participant heard 35 foreground sounds in isolation (none of which were speech nor music), 35 real-world background noises in isolation, and 35 mixtures of a foreground sound and a background noise. In Experiment 3, each participant heard the same 35 foreground sounds as in Experiment 2 (i.e., none of which were speech nor music), 35 mixtures of a foreground sound and a real-world background noise, and 35 mixtures of the foreground sounds and spectrally shaped Gaussian noise (each of which was matched to have the same long-term average spectrum as a real-world background noise). In Experiment 4, each participant heard 108 stimuli total: 54 stimuli during both the auditory and visual tasks, 27 foreground sounds in isolation and 27 mixtures of a foreground sound and a real-world background noise. For each subject, the 27 sounds and noises were a different random subset of the 35 foregrounds and 35 backgrounds used in Experiment 2 and 3 (again, none of which were speech nor music). The total number of stimuli for each subject in Experiment 4 (54) was lower the previous experiments (respectively, 90, 105, and 105), because we doubled the number of presentations of each exemplar, as we had two different task conditions (one visual and one auditory, as described below).

In order to maximize the diversity of pairings between foregrounds and backgrounds, in all four experiments the pairings were randomized across participants—each participant heard each foreground and each background once in a mixture, but one participant might hear a person screaming mixed with a disposal, while another may hear a person screaming mixed with polite applause. The SNR (ratio of power of the foreground versus the background noise) for the four experiments were, respectively: −6 dB, 0 dB, 0 dB, and 0 dB. To encourage attention to each stimulus, participants in Experiments 1, 2, and 3, as well as in the auditory task in Experiment 4, performed a sound intensity discrimination task on the stimuli (described in more detail below).

**Experimental stimuli: spectrally matched noise**. We generated the spectrally shaped Gaussian noise by computing the Fourier amplitude spectrum for each real-world background noise, generating random phases by taking the Fourier transform of a Gaussian vector of the same length as the background noise, and replacing the original phases with these random phases. We generated a new time-domain signal by taking the inverse Fourier transform.

**Experimental stimuli: visual task in Experiment 4**. The visual task in Experiment 4 was a one-back task. Participants were presented a series of 4 × 4 grids, each with six squares filled in. Each grid was on the screen for 695 ms, followed by a 122 ms interstimulus interval during which a circle in the center of the screen was shaded green if participants had given a correct response for the previous stimulus and red if not. Four stimuli were presented successively during each TR (which was 3.27 s in Experiment 4). The location of two of the colored squares changed from stimulus to stimulus unless the pattern repeated. Repeats occurred 30% of the time. Participants had to report these repeats with a button press during the presentation of the stimulus (i.e., within 695 ms). To become familiarized with the task, participants performed a practice run of the visual task before the scan.

During all runs of Experiment 4, participants were presented with both the stream of visual grids and the auditory stimuli (i.e., the visual and auditory stimuli were presented during both the visual and auditory task). The only difference between the stimuli that were presented to the subjects during the two tasks is that the participants did not receive feedback during the auditory task runs (the central visual circle simply turned green or red randomly; turning red with probability 0.1).

**fMRI stimulus presentation**. Sounds were presented using a mini-block design (Supplementary Fig. S2). Each mini-block consisted of three presentations of the identical sound clip. After each presentation, a single fMRI volume was collected, such that sounds were not presented simultaneously with the scanner noise ("sparse scanning"). Foreground sounds were 2-s long, and were presented with 200 ms of silence before and after each sound. Background noises were 2.4-s long. Mixtures were 2.4-s long, with the foreground starting 200 ms after the start of the background and ending 200 ms before the end of the background. Foregrounds and backgrounds had asynchronous onsets because common onsets are a well-established cue to perceptually group sounds[70], which we sought to avoid. In Experiments 1, 2, and 3, each block was 8.88 s (three repetitions of a 2.96 s TR); in Experiment 4, each block was 9.81 s (three repetitions of a 3.27 s TR). This three-presentation block design was selected based on pilot experiments that showed that given the same amount of overall scan time, a three-presentation block gave more reliable BOLD responses than an event-related design or a design with additional repetitions (e.g., with five presentations per block).

For Experiments 1, 2, and 3, blocks were grouped into runs (each run was ~6.5 min), with either thirty (Experiment 1) or 35 (Experiment 2 and 3) stimulus blocks presented in each run. Across three consecutive runs, all stimulus blocks were presented once. Each participant had 12 runs total across a single 2-h scanning session, and thus each stimulus block was presented four times. The order of stimuli was randomized across each set of three runs. For Experiment 4, 27 stimulus blocks were presented in each run (each run was ~6-min long), and all stimuli were presented once across two consecutive runs. The order of the stimuli was randomized across each set of two runs. Each participant had 16 runs in Experiment 4, and thus each auditory stimulus was presented eight times (four times during the visual task and four times during the auditory task). The task was swapped every two runs (i.e., two runs auditory task, two runs visual task, etc.). The task that participants performed first was alternated across subjects.

To enable estimation of the baseline response, silent "null" blocks were included (~20% of TRs), which were the same duration as the stimulus blocks and were randomly interleaved throughout each run. In Experiment 4, neither the visual nor the auditory stimulus was displayed during these silent blocks.

In each block, one of the three presentations was 7 dB lower in level than the other two (the lower-intensity presentation was never the first sound). Participants were instructed to press a button when they heard the lower-intensity stimulus. Participants were instructed to keep their eyes open during the auditory task in all experiments (Experiment 1, 2, 3, and 4). Sounds were presented through MR-compatible earphones (Sensimetrics S14) at 72 dB SPL (65 dB SPL for the quieter sounds). Stimulus presentation code made use of MATLAB and PsychToolbox.

**fMRI acquisition**. MR data were collected on a 3T Siemens Trio scanner with a 32-channel head coil at the Athinoula A. Martinos Imaging Center of the McGovern Institute for Brain Research at MIT. T1-weighted anatomical images were collected in each participant (1-mm isotropic voxels) for alignment and cortical surface reconstruction. In Experiments 1, 2, and 3, each functional volume consisted of 21 slices oriented parallel to the superior temporal plane, covering the portion of the temporal lobe superior to and including the superior temporal sulcus. Repetition time (TR) was 2.96 s (although acquisition time was only 560 ms), echo time (TE) was 30 ms, and flip angle was 90°. In Experiment 4, we expanded the number of slices from 21 to 33, so that we could be sure to acquire the entire occipital lobe in all subjects in order to record BOLD responses in visual cortical areas (i.e., to evaluate mean responses in the visual cortex during the two tasks as we report in Fig. 4b). As a result, the acquisition time was increased to 870 ms, and thus the TR was increased to 3.27 s. All other acquisition parameters were kept the same as they

were in Experiment 1, 2, and 3. For each run in all four experiments, the four initial volumes were discarded to allow homogenization of the magnetic field. In-plane resolution was $2 \times 2$ mm ($96 \times 96$ matrix), and slice thickness was 2.8 mm with a 10% gap, yielding an effective voxel size of $2 \times 2 \times 3.08$ mm.

A simultaneous multislicing (SMS) factor of three was used to expedite the time of acquisition. The factor of three was selected via pilot experiments comparing protocols with different SMS factors (both higher and lower). In these pilots, we sought to maximize the degree to which responses measured from the auditory cortex were reliable across presentations of the same stimulus and differentiated across presentations of different stimuli. We measured the correlation between multi-voxel response patterns to different sounds, as well as to the same sound presented multiple times. We computed the "separability" of the voxel patterns as the difference between the mean test-retest correlation (i.e., reliability) of the pattern and the mean of the correlation between patterns evoked by different pairs of stimuli. An SMS factor of three maximized this quantity of separability. Responses measured with higher SMS factors were more similar across presentations of the same stimuli, but were also more similar across presentations of different stimuli, presumably due to the increase in smoothing induced by higher SMS factors.

**fMRI analysis: preprocessing and response estimation.** Functional volumes were preprocessed with FreeSurfer's FSFAST and in-house MATLAB scripts. Volumes were corrected for motion and skull-stripped. Each run was aligned to the anatomical volume using FSL's FLIRT and FreeSurfer's BBRegister. These preprocessed functional volumes were then resampled to the reconstructed cortical surface. The value for each point on the surface was computed as the average of the (linearly interpolated) value at six points across the cortical ribbon: the pial boundary, the white matter boundary, and four evenly spaced locations between the two. In order to improve SNR, responses were smoothed on the surface with a 3 mm full-width-at-half-maximum (FWHM) 2D Gaussian kernel for the ROI analyses and with a 5-mm kernel for the group summary maps. All analyses were done in this surface space. The elements in this surface space are sometimes referred to as vertices, but for ease of discussion we refer to them as voxels throughout this paper. We used a general linear model (GLM) to estimate the response to each of the stimuli, employing a separate boxcar regressor for each stimulus. We had 90 such regressors for Experiment 1, 105 for Experiments 2 and 3, and 108 for Experiment 4 (the 27 isolated foreground sounds and the 27 mixtures separately estimated in the visual and the auditory task). These boxcars were convolved with a hemodynamic response function, which was assumed, as is often standard, to be a gamma function ($d = 2.25$; $t = 1.25$). Nuisance regressors included a linear and a quadratic regressor to account for drift and three additional regressors to help account for the residual effects of participant motion (the top three PCs of the six translation and rotation motion correction degrees of freedom).

**fMRI analysis: voxel-wise invariance index.** To measure the robustness of each voxel's response to the presence of background noise, we computed the Pearson correlation between the voxel's response to the foreground sounds and to the same foregrounds embedded in background noise (i.e., the mixtures). Different voxels have different levels of measurement noise (e.g., due to distance from the measurement coils), and thus to enable comparisons across voxels we corrected for this measurement noise by employing the correction for attenuation[44]. Our invariance index is the square of this corrected correlation coefficient: an estimate of the (squared) correlation coefficient of that voxel's response to foregrounds and foregrounds embedded in noise that would be measured in the limit of infinite data.

To employ this correction, we measured the response to foregrounds and mixtures separately in even- and odd-numbered presentations (i.e., we averaged responses to the second and fourth and to the first and third presentations in a scanning session). We computed the correlation between responses to foregrounds and mixtures for the even and odd presentations, and took the average of these two correlation coefficients. We averaged the correlation coefficients after applying the Fisher r-to-z transform, and then we returned this mean value the original domain by applying the inverse Fisher z-to-r transform. We averaged in this z-space, because the sampling distribution of correlation coefficients is skewed; averaging in z-space reduces the bias of the estimates of the true mean. We then measured the reliability of the response to the foregrounds and the reliability of the response to the mixtures, by computing the Pearson correlation between the responses to even and odd presentations of the foregrounds and mixtures, respectively. We then applied the correction for attenuation, dividing the correlation of the response to the foregrounds and mixtures by the geometric mean (square root of the product) of the reliability of the responses:

$$R_{\mathrm{corrected}} = \frac{\mathrm{mean}_z \left( \mathrm{r}\left(\mathbf{v}_{fg}^{\mathrm{even}}, \mathbf{v}_{\mathrm{mix}}^{\mathrm{odd}}\right), \mathrm{r}\left(\mathbf{v}_{fg}^{\mathrm{odd}}, \mathbf{v}_{\mathrm{mix}}^{\mathrm{even}}\right) \right)}{\sqrt{\mathrm{r}\left(\mathbf{v}_{fg}^{\mathrm{even}}, \mathbf{v}_{fg}^{\mathrm{odd}}\right) * \mathrm{r}\left(\mathbf{v}_{\mathrm{mix}}^{\mathrm{even}}, \mathbf{v}_{\mathrm{mix}}^{\mathrm{odd}}\right)}}$$

Where $\mathrm{mean}_z$ denotes the average computer after the Fisher r-to-z transform, the r denotes a function that computes the Pearson correlation coefficient, and $\mathbf{v}$ is a d-length vector of the voxel's response to either the foregrounds sounds ("fg") or the

mixtures of foreground sounds and background noise ("mix") from either the even or odd presentations. We square this correlation coefficient to yield a measure of variance explained, which is our invariance index, I:

$$I = R_{\mathrm{corrected}}^2$$

We only included voxels in our analyses that both: (1) met a liberal criterion of sound responsiveness (uncorrected $p < 0.001$, one-sample $t$ test, sound response greater than silence), and (2) exhibited reliable responses both for the foregrounds in isolation and for the mixtures. The reliability criterion used throughout this paper was a Pearson correlation coefficient of 0.2, but similar results were seen when the criterion was made more lenient or conservative (a criterion of 0.01 or 0.40, respectively). See Supplementary Fig. S3 for maps of average reliability. All analyses, including the computation of noise robustness, were performed in individual participants. For the summary maps, we transformed the individual participant maps to Freesurfer's fsaverage space and averaged the robustness measure for each voxel across all participants for whom that voxel met the inclusion criteria (i.e., with a sufficiently reliable response). We did so for all voxels where at least three participants exhibited reliable responses.

**fMRI analysis: regions of interest.** For regions of interest (ROI) analyses, we used a primary ROI defined as TE 1.1 and 1.0[10] and a non-primary ROI defined as the anterior and posterior superior temporal gyrus parcels from Freesurfer.

**fMRI analysis: multi-voxel pattern-based invariance measures.** In addition to examining the robustness of responses of individual voxels across sounds, we also examined the robustness of the response pattern across voxels to a given foreground sound. If the response in a region is robust to the presence of a background noise, then the pattern across voxels within that region should be similar when a sound is presented in isolation or embedded in background noise. We analyzed the data from Experiment 3 and performed a pattern-based decoding analysis, using a nearest neighbor classifier with Pearson correlation as the distance metric. We demeaned each voxel within an ROI (i.e., subtracted its mean response across sounds, as is standard in multi-voxel pattern analysis). We then correlated the response across voxels to the $i$th foreground in isolation with the response to each of the foreground–background mixtures. If the maximal correlation was with the mixture containing the $i$th foreground, we declared the classification correct; otherwise we declared it incorrect. We performed this analysis in the primary and non-primary ROIs separately, doing so in each participant for both the real-world and synthetic background noises. The results are shown in Supplementary Fig. S8A.

One potential pitfall of such a decoding analysis is that it could be contaminated by reliability differences between the ROIs. As an alternative, we compared the pattern of activity across voxels in the two ROIs to a given sound in isolation with the pattern evoked by the same sound embedded in noise. For consistency with the voxel-wise invariance metric that we use throughout the paper, we used the correlation between the two response patterns as the measure of similarity. Critically, we normalized the correlation by maximum possible value given the reliabilities of the two response patterns (analogous to the voxel-wise metric). As with the voxel-wise invariance metric, if the pattern of responses across voxels is robust to the presence of background noise, the resulting correlation coefficient should be high. The results are shown in Supplementary Fig. S8B.

**fMRI analysis: mean percent signal change in ROIs.** For the mean response plots in Experiment 4 (Fig. 4b), we selected all voxels that responded significantly to either the visual or auditory stimulus ($p < 0.001$, uncorrected, stimulus versus silence) and that were in the relevant region of interest. The auditory cortex ROI was the union of the primary and non-primary auditory cortical ROI used above. The visual cortex ROI was the PALS atlas occipital lobe parcel. For the mean responses to different auditory stimuli (Supplementary Figs. S5, S9, and S10), we used the same primary and non-primary ROIs as used in the main text, and again selected voxels based on their responsiveness to sounds ($p < 0.001$, uncorrected sounds versus silence).

**Sample sizes.** Pilot versions of Experiment 1 indicated that the effect size of the difference in the mean of the invariance metric across voxels in each ROI was large. Indeed, there was a 90% chance of rejecting the null hypothesis that each participant's mean invariance in each ROI was are equal with just two participants, as evaluated with a paired two-tailed $t$ test with a $p$-value criterion (alpha) of 0.05. However, we also wanted to examine the invariance metric in maps across all of auditory cortex, and we wanted those maps to be reliable. Using pilot data (from the first four participants in Experiment 1), we estimated the split-half reliability of the maps (i.e., the correlation of the two maps derived from splits of the participant set) as a function of the number of participants using the Spearman–Brown correction. The predicted reliability ranged between $r = 0.36$ ($n = 2$) to $r = 0.89$ ($n = 30$). We chose a sample size of twelve participants, which yielded a projected reliability of $r > 0.75$, which we considered reasonable.

**Reporting summary**. Further information on research design is available in the Nature Research Reporting Summary linked to this article.

## Data availability
All data are available upon request.

## Code availability
All code is available upon request.

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

## Acknowledgements

The authors thank Erica Shook for assistance with collecting the fMRI data, Steve Shannon for MR support, Atsushi Takahashi for assistance with MR acquisition protocol design, and Satra Ghosh and the OpenMind team for computing resources. The authors also thank Dana Boebinger, Sam Norman-Haignere, Erica Shook, and Pedro Tsividis for comments on the paper. Work supported by a DOE Computational Science Graduate Fellowship (DE-FG02-97ER25308) to A.K., an NIH NRSA Fellowship (F32DC017628) to A.K., a McDonnell Scholar Award to J.H.M., NIH grants 1R01DC014739-01A1 and R25GM067110, and NSF grants BCS-1634050 and PHY17-48958.

## Author contributions

A.J.E.K. and J.H.M. designed the research. A.J.E.K. collected the data, analyzed the data, and wrote the first draft of the paper. A.J.E.K. and J.H.M. reviewed and edited the paper.

## Additional information

**Competing interests:** The authors declare no competing interests.

