## [Peer Review File · Nature Communications]

Reviewers' Comments:

Reviewer #1:

Remarks to the Author:

This fMRI study reports data from four experiments exploring auditory cortical responses to mixtures of target sounds embedded in background noise. The basic finding, nicely replicated across the different conditions, is that a measure of consistency of response to the target sounds is much less affected by the presence of background noise in nonprimary areas of the temporal lobe than in peri-primary auditory cortex. Thus, the authors conclude, primary and nonprimary regions may be distinguished by their differential invariance to sound properties of target sounds embedded in noise.

The findings are really very clear and convincing in terms of the anatomical segregation. The method used is interesting and novel in some respects, notably in demonstrating that the principal result obtains much more clearly for stationary background sounds, which are more akin to real-world noise, than for synthetic noise as used in most of the literature. The authors also demonstrate that the nature of the foreground sounds (whether they contain speech and music or not) does not alter the result. Finally, they show that even during performance of a demanding visual task, the segregation is maintained, suggesting that the differential processing in the two regions is not dependent upon active attention to the target stimulus.

The authors should be congratulated for a clear and programmatic set of experiments that converge on a clear set of results. It is rare to see several separate experiments in one paper, which adds to the clarity and convincingness of the findings (I also liked that they show that it is possible to obtain perfectly good clear results with sample sizes of 11-12 individuals, despite the sometimes arbitrary demand in neuroimaging these days for much larger sample sizes).

Despite its many good points my enthusiasm for the paper is dampened by a few points.

1. Conceptually, I am not certain how much of a major advance this study represents over others. As the authors are careful to point out, several other older papers have come to similar conclusions, albeit more limited because they either used only speech sounds as targets, and/or because they used less realistic synthetic noise as background. It is useful to show that the phenomenon of nonprimary auditory cortical invariance is not confined to speech, but that seems more like an important detail rather than a breakthrough. Regarding the noise, the authors show that the anatomical distinction is less clear when using synthetic noise; but if prior studies using such noise also found a similar result despite using such nonrealistic stimuli, it argues that it must in fact be a fairly robust effect. Either way, the current finding would represent an extension and clarification of prior conclusions. An advance in our understanding, no doubt, but not certain of its broader significance.

2. The measure used to index invariance in cortical response is interesting. My expertise in fMRI analysis is insufficient to judge its validity or how it compares to other approaches. But I might have liked to see an analysis demonstrating the degree to which the voxel correlation index used here might differ across cortical areas. For instance, it would be useful to look at the correlation to the same stimulus presented on multiple occasions without noise, to establish a baseline of invariance, independently of the presence of background noise. One would expect the measure to be very similar across anatomical regions if the authors' interpretation is correct, that the invariance emerges in the presence of the noise.

3. Still thinking about the invariance index, I wonder why the authors did not take what would appear to be a more obvious and robust approach, using classification. As I read the paper I expected the analysis to unfold in the following manner: first, determine the classification accuracy for target

sounds in each region in the absence of any noise. Then, add noise parametrically and show that classification is degraded in primary areas, but remains high in nonprimary areas. This could be done using the classifiers trained in the no-noise condition, which would be an elegant way (it seems to me at least) to show the robustness of nonprimary regions to capturing the acoustical elements that are relevant to auditory object processing.

4. The attentional manipulation is interesting, and adds further important evidence to the findings. However, I am not sure if it addresses the issue of how attention might differ for the foreground and background sounds. The visual/auditory manipulation demonstrates that the nonprimary invariance is still present even when attention is directed to another modality, which suppresses overall activity in auditory cortex somewhat (not that much actually, judging from Fig 4B). OK, that is convincing; but it does not address whether any remaining attentional resources are captured differently by the properties of the foreground and background sounds. To do that, one would have to manipulate attention *within* the auditory modality, asking listeners to attend to some property of the background vs some property of the foreground (for instance). The task used in all but the last experiment to control for attention (detect a loudness change in one of the three stimulus presentations within a miniblock) is adequate to maintain alertness, but it is not a selective attention task, since the loudness manipulation was applied to the entire stimulus, both target and embedded noise, if I understood the methods correctly. So in general, the issue of attention and what role it may play in the findings is not as clearly resolved as one might desire (in fairness, it is a complicated issue and no doubt would require another set of studies).

Minor point: It would be useful to know what the field of view was for this study, given that it was restricted to the temporal lobe.

Reviewer #2:

Remarks to the Author:

This is an interesting and clearly written article describing an fMRI study comparing the relative variability of responses to different natural foreground sounds when they are presented in isolation versus when played at the same time as a background scene (with foreground/background sounds characterized using acoustic analyses). The results - that fMRI responses to sounds in 'non-primary' auditory regions are more invariant than in more 'primary' regions - were shown across four different experiments that varied in sound category (with speech & music sounds included or not) and attentional demands (with the pattern of differential invariance quite unaffected by a concurrent 1-back visual task).

The findings are interesting, well and carefully couched in the literature, and show a compelling difference in the 'style' with which auditory regions respond to sounds in different acoustic environments, even when attentional demands are quite different. I suspect that other groups might try this approach to discover relationships between different functional topographic features and relative invariance under different acoustic and task conditions. The robustness of these 'invariance maps' to the inclusion/exclusion of speech and music stimuli was particularly interesting, given how suggestively similar the map of the shift in invariance along the STG is to (for instance) comparisons of speech and non-speech stimuli.

The work is also very carefully carried out, and gives me confidence in the robustness of the results. I am enthusiastic about this study, and think it will definitely provoke interest in the larger community. The manuscript is exceptionally clean and well structured, and I have really only one substantive comment, and notes on a few minor details.

Lines 132-139, regarding the difference between the relative invariance of activations to sound in 'primary' and 'non-primary' areas being less much robust with synthetic versus real-world backgrounds. I agree entirely that there is an obvious difference between the two background conditions, but actually think there is an interesting finding in Experiment 3 that is getting hidden by the very generous 'primary' label used (the full extent of the probabilistic TE1.0 and 1.1). It is hard to tell from the figure since the gyral anatomy is hidden by the statistics map, but it looks to me as though the less-invariant patch in both hemispheres (but particularly the left) is where one would predict primary auditory cortex to be on average, e.g., the medial 2/3rds of Heschl's. If true, this would suggest that the authors might have found a rather sensitive index of primary auditory areas (~A1/R). If so inspired, the authors could draw an anatomical patch in this more constrained region and compare the invariance values to lateral STG (for example).

Details/Minor

Fig 1 - it would be helpful if possible to have a schematic with as much information about fMRI paradigm/acquisition as possible if it can be fit in. (Or even a new figure if the journal doesn't consider it too profligate).

Methods: I may have missed it but did participants in Experiments 1-3 close their eyes, or look at the screen (or something else)?

line 591 '... analyses were done in this surface space, but for ease of discussion we refer to vertices as "voxels" throughout': this is actually more confusing since values from multiple (actual) voxels will contribute to a single vertex.

Reviewer #3:

Remarks to the Author:

The study by Kell & McDermott investigates how the auditory cortex transforms a representation of the actual sound stimulus to an invariant representation of the "foreground" sound. It is suggested that primary auditory cortex is largely invariant to simple artificial "background" sounds, while only nonprimary auditory cortex is invariant to real-world "background" sounds. The results demonstrate a hierarchy of processing stages towards a invariant, object-based auditory representation. The results are solid and the manuscript is also pleasant to read. I have a few concerns, however, about the interpretation of the data.

1. The authors showed that the neural representations of foreground sounds are invariant to background sounds. The foreground and background sounds are defined based on acoustic stationarity, i.e., whether the statistical property changes over time. This definition, however, is somewhat arbitrary. It is certainly true that background sounds are usually stationary while foreground sounds are more dynamic, but I'm not sure if stationarity is the defining feature of foreground/background sounds. Maybe this definition is supported by the current results but it seems a bit arbitrary to use it as a priori. Furthermore, people can choose to listen to a stationary sound, e.g., rain sound or wind sound.

What the current results show is that, when stationary and nonstationary sounds are simultaneously presented, nonprimary auditory cortex selectively responds to the nonstationary component. Based on this interesting result, I think a more appropriate conclusion is that nonprimary auditory cortex selectively represents nonstationary sounds. Related to this point, I wonder if the authors have data

about how nonprimary auditory cortex responds to independently presented nonstationary sounds. I wonder if the lack of responses to nonstationary sounds is due to feature tuning properties in nonprimary auditory cortex, i.e., the lack of neurons sensitive to stationary sounds, or due to a competition between sound sources.

2. The authors showed a nice hierarchy of processing stages for stationary sounds, simple stationary sounds, and realistic stationary sounds. I wonder if this hierarchy is related to the hierarchy of temporal windows in different auditory processing stages. We know that low-level auditory processing stages can phase lock to very fast temporal modulations while the phase-locking ability reduces along the ascending pathway. My guess is that the synthetic stationary sounds are stationary (based on the working definition in this paper) across very short time windows (e.g., 10 ms), while the real-life stationary sounds are only stationary for longer time windows. If this is true, the selectivity to nonstationary sound may be attributed to the longer temporal integration window in nonprimary auditory cortex. Related to this point, it will be useful to show the modulation spectrum of different sets of stimuli in the supplementary figures.

Minor points:

1. Please show the boundary of the scanned area and the boundary of nonprimary auditory cortex in the figures.

2. Experiment 2 is done to exclude speech and music. I wonder why this is necessary. They could be excluded based on the data from Experiment 1. Is it because Experiment 1 has too many speech/music sounds?

3. Line 170, how about nonprimary auditory cortex? Are the responses changed by the task?

4. For the short time windows used in the current study, it's not meaningful to discuss very low-frequency modulations, e.g., 0.5 Hz modulations (Line 438).

Response to the Editor and Reviewers

We thank the reviewers for the careful read and the helpful criticism. We have spent the past months conducting an extensive set of additional analyses and experiments, and believe the paper is much improved. These analyses are evident in eight new supplementary figures, numerous text revisions, and the responses to the reviews below. We hope the paper is now considered suitable for publication.

Response to the Reviewer 1

Reviewer #1 (Remarks to the Author):

This fMRI study reports data from four experiments exploring auditory cortical responses to mixtures of target sounds embedded in background noise. The basic finding, nicely replicated across the different conditions, is that a measure of consistency of response to the target sounds is much less affected by the presence of background noise in nonprimary areas of the temporal lobe than in peri-primary auditory cortex. Thus, the authors conclude, primary and nonprimary regions may be distinguished by their differential invariance to sound properties of target sounds embedded in noise.

The findings are really very clear and convincing in terms of the anatomical segregation. The method used is interesting and novel in some respects, notably in demonstrating that the principal result obtains much more clearly for stationary background sounds, which are more akin to real-world noise, than for synthetic noise as used in most of the literature. The authors also demonstrate that the nature of the foreground sounds (whether they contain speech and music or not) does not alter the result. Finally, they show that even during performance of a demanding visual task, the segregation is maintained, suggesting that the differential processing in the two regions is not dependent upon active attention to the target stimulus.

The authors should be congratulated for a clear and programmatic set of experiments that converge on a clear set of results. It is rare to see several separate experiments in one paper, which adds to the clarity and convincingness of the findings (I also liked that they show that it is possible to obtain perfectly good clear results with sample sizes of 11-12 individuals, despite the sometimes arbitrary demand in neuroimaging these days for much larger sample sizes).

Thank you.

Despite its many good points my enthusiasm for the paper is dampened by a few points.

1. Conceptually, I am not certain how much of a major advance this study represents over others. As the authors are careful to point out, several other older papers have come to similar conclusions, albeit more limited because they either used only speech sounds as targets, and/or because they used less realistic synthetic noise as background. It is useful to show that the phenomenon of nonprimary auditory cortical invariance is not confined to speech, but that seems more like an important detail rather than a breakthrough. Regarding the noise, the authors show that the anatomical distinction is less clear when using synthetic noise; but if prior studies using such noise also found a similar result despite using such nonrealistic stimuli, it argues that it must in fact be a fairly robust effect. Either way, the current finding would represent an extension and clarification of prior conclusions. An advance in our understanding, no doubt, but not certain of its broader significance.

We are not sure what the reviewer is referring to as a “similar result” in prior findings using synthetic noise. Our reading of the literature is that no one has previously assessed noise invariance across the cortex, regardless of the type of noise. Moreover, no one has demonstrated behaviorally relevant functional differentiation across distinct hierarchical stages (with the exception of speech-related processing, which many believe to occur in a specialized pathway). We view the key advance of our work as using noise invariance to provide evidence for a domain-general representational transformation between primary and non-primary auditory cortex. It was critical in this respect to use realistic sources of noise.

In addition to addressing the fundamental question of the representational consequences of hierarchical organization, our work clarifies the basis of listening in noise – one of the central problems facing the auditory system, and itself the subject of intense recent interest. Part of the interest is due to widespread hearing

disorders in which listening in noise is particularly compromised, and the effect we describe could help to clarify the basis of listening difficulties in impaired populations. Our findings thus seem likely to have relevance to a broad swath of researchers.

A third key contribution of our work is methodological: we introduce a simple and broadly applicable means to assess the invariance of neural representations. As Reviewer 2 notes, researchers can apply the approach we introduce here—measuring the similarity of brain responses between the same source signals in different conditions—to other problems of invariance/robustness in the auditory system (such as reverberation or intensity), as well as to analogous problems of invariance in other sensory systems and beyond.

To emphasize and to clarify what we believe is the key contribution of this manuscript, we have revised the introduction and the discussion. The revised passages read:

“Previous attempts to characterize such transformations have largely been limited to speech^{2, 14-19}, which may involve a specialized pathway^{15, 20, 21}, or have suggested that non-primary areas are more influenced by task or attention^{4, 16, 22}. Here we probed for a general sensory transformation that might differentiate stages of representation, measuring the invariance of sound-evoked responses throughout auditory cortex to the presence of background noise.” (Page 2, Lines 30-34)

And:

“More broadly, these findings reveal a general transformation between auditory cortical stages, illustrating a representational consequence of the putative hierarchical organization in the auditory system that has clear relevance to everyday perception and behavior.” (Page 2, Lines 66-68)

And:

“These results demonstrate that noise invariance may be a generic functional signature of non-primary auditory cortex, illustrating a representational consequence of its putative hierarchical organization with clear relevance to perception and to behavior.” (Page 9, Lines 216-218)

We hope the revision makes the contribution of the work more apparent.

2. The measure used to index invariance in cortical response is interesting. My expertise in fMRI analysis is insufficient to judge its validity or how it compares to other approaches. But I might have liked to see an analysis demonstrating the degree to which the voxel correlation index used here might differ across cortical areas. For instance, it would be useful to look at the correlation to the same stimulus presented on multiple occasions without noise, to establish a baseline of invariance, independently of the presence of background noise. One would expect the measure to be very similar across anatomical regions if the authors' interpretation is correct, that the invariance emerges in the presence of the noise.

We were concerned about precisely this issue, and that is why the invariance metric that we use throughout the paper corrects for differences in measurement noise across voxels. The numerator of the invariance index is simply the correlation between a voxel's response to sounds and sounds embedded in noise. The denominator employs the correction for attenuation (Spearman, 1904) to correct this “raw” correlation coefficient by the test-retest reliability of the voxel's response. The resulting corrected correlation coefficient is an estimate of the correlation coefficient of that voxel's response to sounds and sounds in noise that would be measured in the limit of infinite data. This correction ensures that the ceiling of the invariance index is the same (i.e., 1.0) across voxels. We have added language to further clarify and emphasize this point in the main text:

“If the neurons sampled by a voxel are robust to background noise, this correlation coefficient should be high. Different voxels exhibit different amounts of measurement noise, due to a variety of factors including differences in distance from the receiver coils (Supplementary Fig. S3). To enable comparisons across voxels we corrected for such measurement noise^{39, 40} (see Methods for details). As a result of this reliability correction, any differences in the robustness metric across cortical areas is not the result of differences in measurement noise across cortical areas.” (Page 4, Lines 95-100)

We have also expanded our discussion of this correction in the methods, adding the following:

“Our invariance index is this corrected correlation coefficient: an estimate of the correlation coefficient of that voxel’s response to foregrounds and foregrounds embedded in noise that would be measured in the limit of infinite data.” (Page 23, Lines 662-664)

Additionally, in response to this reviewer’s suggestion, we have generated maps plotting the test-retest reliability of responses across the cortical sheet. These maps have been included in the revised manuscript as Supplementary Figure S3, and we have reproduced them below.

The maps show that reliability is overall lower in primary auditory cortex than non-primary auditory cortex, though there is substantial variation within each ROI. This variation in reliability is roughly what would be expected given that primary auditory cortex is inside a sulcus and thus more distant from the MRI receiver coil than non-primary auditory cortex, much of which is on the superior temporal gyrus, right next to the coil. But these differences in reliability underscore the importance of correcting for reliability, as we did in our noise robustness index. They also show that there are parts of non-primary auditory cortex where reliability is rather low, but whose responses are still quite robust to stimulus noise (e.g., in more lateral and posterior portions).

Figure S3. Reliability of cortical responses.

Figure S3. Reliability of cortical responses.

- (A) Map of the test-retest reliability of voxel responses to natural sounds in isolation from Experiment 1. Reliability was computed in individual subjects, and then averaged across all eleven subjects. Only voxels with an average correlation coefficient greater than 0.2 are assigned a color. For reference, the white outline indicates the primary auditory cortical region of interest (TE 1.1 and 1.0).
- (B) Map of the test-retest reliability of responses to foreground-background mixtures. Plotting conventions, including color scale, are the same as (A).
- (C) Map of the maximum possible correlation (i.e., the noise ceiling) between responses to foregrounds sounds and foreground-background mixtures, which is simply the geometric mean of the values shown in (A) and (B). Plotting conventions, including color scale, are the same as (A) and (B).

3. Still thinking about the invariance index, I wonder why the authors did not take what would appear to be a more obvious and robust approach, using classification. As I read the paper I expected the analysis to unfold in the following manner: first, determine the classification accuracy for target sounds in each region in the absence of any noise. Then, add noise parametrically and show that classification is degraded in primary areas, but remains high in nonprimary areas. This could be done using the classifiers trained in the no-noise condition, which would be an elegant way (it seems to me at least) to show the robustness of nonprimary regions to capturing the acoustical elements that are relevant to auditory object processing.

The main difficulty of using classification in this setting is that it is not obvious how to control for the differences in reliability across ROIs, which could produce differences in classification for reasons unrelated to the underlying hypothesis about the invariance of the representation. This is the main reason we avoided classification in the initial manuscript. In response to this comment, we provide the analysis below and in Supplementary Figure S7, using the voxels from the same ROIs in the paper. The results qualitatively replicate the trends evident from our voxel-wise invariance metric – classification of noisy foregrounds is better in the non-primary ROI, but only for real-world noise sources.

A Classification of presented natural sound (Train on sound in isolation; test in noise)

(A) Proportion of stimuli correctly classified from multivoxel patterns of natural sounds from Experiment 3. Classifiers were trained on foreground sounds in isolation and tested on foreground sounds in noise (either real-world or synthetic), separately for primary and non-primary regions of interest. Dashed gray line indicates chance performance in the 35-way classification. Classification was significantly better in non-primary areas than primary areas for real-world noise ($t_{22} = 2.42$, $p = 0.0244$), but not synthetic noise ($t_{22} = 0.0286$, $p = 0.977$). Error bars plot within-subject SEMs.

To obtain the above numbers, we used a nearest neighbor classifier with Pearson correlation as the distance metric. We analyzed data from Experiment 3 (with real-world and synthetic background noise). We demeaned each voxel within an ROI (i.e., subtracted its mean response across sounds, as is standard in multi-voxel

pattern analysis). We then correlated the response across voxels to the i -th foreground with the response to each of the foreground-background mixtures. If the maximal correlation was with the mixture containing the i -th foreground, we declared the classification correct; otherwise we declared it incorrect. We did this for each ROI separately, in each participant. The numbers plotted in the figure above are means across participants. Two-sample t tests on the classification results across subjects show a significant difference between ROIs for real-world noise ($p=0.024$), but no difference between ROIs for synthetic noise ($p=0.98$).

The trouble with this result is that it could potentially be contaminated by reliability differences between the ROIs. To complement the above results, we ran a decoding analysis for each of the two ROIs in which we compared the pattern of activity across voxels to a given sound in isolation with the pattern evoked by the same sound embedded in noise. For consistency with the voxel-wise invariance metric that we use throughout the paper, we used the correlation between the two response patterns as the measure of similarity. Critically, we normalized the correlation by maximum possible value given the reliabilities of the two response patterns (analogous to the voxel-wise metric). As with the voxel-wise invariance metric, if the pattern of responses across voxels is robust to the presence of background noise, the resulting correlation coefficient should be high.

We evaluated this pattern-based invariance metric in the primary and non-primary regions of interest for Experiment 3, comparing the robustness to real-world noise as well as the synthetic, spectrally matched noise:

B Robustness of pattern across voxels to real-world & synthetic noise

(B) Invariance of the pattern of responses in primary and non-primary regions to real-world and synthetic noise. The values are normalized by the maximum possible value given the reliability of the patterns. Error bars plot within-subject SEMs.

We found a similar set of results with this pattern-based metric as we did with the voxel-based metric. Patterns of voxel responses in primary areas were substantially more robust to the synthetic noise than the real-world noise, while patterns in non-primary areas are largely robust to both types of noise. We have included both of these results in Supplementary Figure S7.

We now mention this analysis in the Results section:

“We obtained qualitatively similar results when we examine the robustness of patterns of responses across voxels to the presence of the two types of background noise. Patterns across voxels in non-primary areas exhibited greater invariance than those in primary areas, both when using the pattern to classify foreground sounds in noise, and when measuring the similarity of the pattern elicited by noisy foregrounds to clear foregrounds (Supplementary Fig. S7).” (Page 6, Lines 160-164)

*4. The attentional manipulation is interesting, and adds further important evidence to the findings. However, I am not sure if it addresses the issue of how attention might differ for the foreground and background sounds. The visual/auditory manipulation demonstrates that the nonprimary invariance is still present even when attention is directed to another modality, which suppresses overall activity in auditory cortex somewhat (not that much actually, judging from Fig 4B). OK, that is convincing; but it does not address whether any remaining attentional resources are captured differently by the properties of the foreground and background sounds. To do that, one would have to manipulate attention *within* the auditory modality, asking listeners to attend to some property of the background vs some property of the foreground (for instance). The task used in all but the last experiment to control for attention (detect a loudness change in one of the three stimulus presentations within a miniblock) is adequate to maintain alertness, but it is not a selective attention task, since the loudness manipulation was applied to the entire stimulus, both target and embedded noise, if I understood the methods correctly. So in general, the issue of attention and what role it may play in the findings is not as clearly resolved as one might desire (in fairness, it is a complicated issue and no doubt would require another set of studies).*

We agree with the reviewer that it would be interesting to manipulate attention within the auditory modality and measure the effect on the invariance of the cortical responses. For instance, participants could be asked to direct their attention to either the “foreground” or “background” sound on a given trial, and we could in principle measure whether our invariance index is altered. We considered this sort of experiment in the early stages of our experimental design, but decided to use the across-modality manipulation that is in the manuscript because we thought it would be difficult to manipulate attention to the foreground and background sounds in a way that we could verify as balanced (e.g. with the same “amount” of attention in the two conditions).

Motivated by the reviewer’s comment, in preparing this revision we made a concerted effort to make such a manipulation work. These efforts confirmed our initial fear: given the constraints of using naturalistic foreground and background sounds, it proved impossible (in our hands) to direct attention to the two types of sounds in a way that would enable a tight experiment.

This is what we tried:

The goal was to come up with a behavioral task that would force subjects to attend to either the foreground of the background. This would seem to necessitate having them make a judgment about one of the two stimuli, and for this judgment to be similar across stimuli. The auditory task that we used in the cross-modal attentional experiment required participants to discriminate the intensity of successive sounds, and we tried to adapt this to force participants to pay attention to one sound or the other. We thus designed a task where participants heard three sounds, and had to judge whether the second or third instance was different in level. There were four conditions:

- 1) foregrounds alone
- 2) backgrounds alone
- 3) mixture of foreground and background, attend foreground
- 4) mixture of foreground and background, attend background

In the mixture conditions, the foreground and background were each different in level in the second or third interval, independently selected (such that on half of trials the foreground and background level increment were in the same interval, and on half of trials they were in different intervals). The subject was cued before the trial to attend to either the foreground or background, with a verbal phrase identifying the sound source (e.g. “car accelerating” or “applause”).

We needed to verify that attention was successfully manipulated by this task. In the attentional experiment described in the manuscript (Experiment 4), we employed visual and auditory tasks to direct attention to one modality or the other. Due to the cross-modal nature of the experiment, we could verify that the tasks successfully altered attention by examining mean responses in visual or auditory cortex. Mean responses in

visual cortex should be higher when the participant is doing the visual task and vice versa for the auditory task, and they were. Because the result of Experiment 4 in terms of cortical robustness was a null effect (inattention did not alter the robustness), having independent verification that the designed attentional manipulation in fact drove attention was critical to interpreting the results.

Given that we anticipated a similar null result when directing attention to one of the two concurrent sources, some way of independently validating the attention manipulation again seemed critical. For this within-modality experiment, it was not obvious what brain-based signature we could use to verify that attention was successfully manipulated, and we therefore turned to behavior to verify that attention was successfully manipulated.

Previous work in the lab had used a “vibrato” detection task for this purpose, where listeners detected small f_0 modulations applied to one of two concurrent synthetic voices (Woods & McDermott, 2015). We thought amplitude modulations would be the best choice of perturbation for arbitrary natural sounds, as they can be imposed multiplicatively on the waveform. We selected a modulation rate that could be detected about equally well when applied to the foreground and background sounds (8 Hz), and modulation depths that were detectable, but at sub-ceiling levels of performance.

The participants were instructed to discriminate the intensity of the cued sound, and to additionally report modulation on either sound source. In our previous paper (Woods & McDermott, 2015) we found that when participants were instructed to attentively track one of two voices, they were better at detecting vibrato when it occurred in the attended voice. This figure is reproduced below for convenience:

Figure 3. Experiment 2: Vibrato Detection as a Measure of Attention during Streaming

(A) Example stimulus trajectories. Either voice could contain vibrato (a brief pitch modulation, added in this example to the green trajectory). Listeners performed the stream-segregation task from experiment 1 but were additionally asked to detect vibrato in either stream. The trajectory shown is 2 s in duration (from experiment 2A); trajectories in experiment 2B were 3 s. (B) Stream-segregation performance for the 12 participants in experiment 2A. (C) Sensitivity to vibrato in the cued and uncued voices for subjects grouped by streaming performance (into two equal-sized groups; left) and pooled across groups (right). Includes only trials in which the stream-segregation task was performed correctly. Error bars here and elsewhere denote within-subject SEMs and thus do not reflect the variability in overall vibrato detection across subjects. (D) Stream-segregation performance for the six best streamers in experiment 2B (3 s mixtures, 250 ms cue and probe, different group of listeners). (E) Sensitivity to vibrato versus temporal position of vibrato onset (equal-sized bins of uniformly distributed onset times) in the cued and uncued voices for the six best streamers in experiment 2B. Only trials in which the stream-segregation task was performed correctly are included. The gray bar below depicts the time course of the mixture; regions matching the cue and probe are in dark gray.

We hoped that the cued intensity discrimination task would produce a similar result with modulation detection (better performance for the cued source). Data from a pilot behavioral experiment with 9 participants is shown below. The experiment contained only the critical mixture conditions (conditions 3 and 4 from the list above). We managed to approximately equate modulation detection for the foreground and background sounds, but there was no obvious effect of attention on modulation detection.

Modulation detection performance (d') by condition, plotted separately for 8 dB (left) and 5 dB modulation depths (right). Error bars plot SEM across the nine participants. Orange bars indicate conditions where attention was directed towards the stimulus that was modulated. If attention was successfully directed by the attentional task, the orange bars should be higher than the gray bars.

We tried several variants of this experiment, including varying the difficulty by changing the modulation depth and rate and selecting the most salient modulation rate for each stimulus individually. In all we ran 6 different variants of this kind of experiment with 33 total participants, but we never saw differences in modulation detection as a function of whether the source was cued or not.

We suspect this is because the intensity task can be performed to some extent by listening for the relative levels of the sounds, such that listeners do not have to focus exclusively on the cued sound. And it is not obvious (to us) what other task could be used with a diverse set of natural sounds while simultaneously allowing a way to measure whether attention was successfully manipulated.

These pilot experiments substantiate our earlier intuition that it is difficult to force attention to the foreground or background sounds in a way that can be verified given a diverse set of natural sounds. We have thus opted not to perform a new fMRI experiment or to include the pilot behavioral results in the paper, but provide the results here to help explain our choices.

Minor point: It would be useful to know what the field of view was for this study, given that it was restricted to the temporal lobe.

We agree, and have included a heatmap of the acquisition window in Figure 1D, which we reproduce below:

Reviewer #2 (Remarks to the Author):

This is an interesting and clearly written article describing an fMRI study comparing the relative variability of responses to different natural foreground sounds when they are presented in isolation versus when played at the same time as a background scene (with foreground/background sounds characterized using acoustic analyses). The results - that fMRI responses to sounds in 'non-primary' auditory regions are more invariant than in more 'primary' regions - were shown across four different experiments that varied in sound category (with

speech & music sounds included or not) and attentional demands (with the pattern of differential invariance quite unaffected by a concurrent 1-back visual task).

The findings are interesting, well and carefully couched in the literature, and show a compelling difference in the 'style' with which auditory regions respond to sounds in different acoustic environments, even when attentional demands are quite different. I suspect that other groups might try this approach to discover relationships between different functional topographic features and relative invariance under different acoustic and task conditions. The robustness of these 'invariance maps' to the inclusion/exclusion of speech and music stimuli was particularly interesting, given how suggestively similar the map of the shift in invariance along the STG is to (for instance) comparisons of speech and non-speech stimuli.

The work is also very carefully carried out, and gives me confidence in the robustness of the results. I am enthusiastic about this study, and think it will definitely provoke interest in the larger community. The manuscript is exceptionally clean and well structured, and I have really only one substantive comment, and notes on a few minor details.

Thank you.

Lines 132-139, regarding the difference between the relative invariance of activations to sound in 'primary' and 'non-primary' areas being less much robust with synthetic versus real-world backgrounds. I agree entirely that there is an obvious difference between the two background conditions, but actually think there is an interesting finding in Experiment 3 that is getting hidden by the very generous 'primary' label used (the full extent of the probabilistic TE1.0 and 1.1). It is hard to tell from the figure since the gyral anatomy is hidden by the statistics map, but it looks to me as though the less-invariant patch in both hemispheres (but particularly the left) is where one would predict primary auditory cortex to be on average, e.g., the medial 2/3rds of Heschl's. If true, this would suggest that the authors might have found a rather sensitive index of primary auditory areas (~A1/R). If so inspired, the authors could draw an anatomical patch in this more constrained region and compare the invariance values to lateral STG (for example).

We thank the reviewer for this suggestion. We have performed this analysis, and have reproduced it below, as well as included it as Supplementary Figure S6. There is no obvious difference between the different regions of primary auditory cortex, at least given the available anatomical ROIs.

We analyzed the original TE1.1 and TE1.0 ROIs as well as those suggested by the reviewer: the conjunction of Heschl's gyrus and the TE ROIs. The gyrus was defined with freesurfer's cortical curvature. We included a step of eroding and dilating the conjunction to get rid of straggler vertices -- e.g., it is evident that the 1.1 ROI would include a small anterior lip of HG. The eroding/dilating step excluded that.

Top: Mean invariance index in regions of interest in and around Heschl's gyrus (HG) from Experiment 3. From left to right: TE 1.1, TE 1.0, the portion of TE 1.1 within Heschl's gyrus, and the portion of TE 1.0 within Heschl's gyrus. Height of point indicates within-subject SEMs. The invariance index was not significantly different between TE 1.1 and 1.0 (real-world noise: $t_{18} = 0.66$, $p = 0.518$; synthetic noise: $t_{18} = 0.47$, $p = 0.689$) or the sub-regions constrained to lie within Heschl's gyrus (real-world noise: $t_{18} = 1.39$, $p = 0.182$; synthetic noise: $t_{18} = 1.22$, $p = 0.240$). Bottom left: Lateral view of inflated brain with ROIs. Bottom right: Zoomed-in view of ROIs.

This analysis is described in the results section:

“Although there is some variation in the invariance metric within the primary ROI, medial and lateral ROIs within Heschl's gyrus corresponding to the TE 1.1 and 1.0 regions showed similar mean values of the metric, indicating some consistency across traditional divisions of primary auditory cortex (Supplementary Fig. S6).” (Page 6, Lines 155-158)

Details/Minor

Fig 1 - it would be helpful if possible to have a schematic with as much information about fMRI paradigm/acquisition as possible if it can be fit in. (Or even a new figure if the journal doesn't consider it too profligate).

We have included a new supplementary figure (S2 in the revised manuscript) showing a schematic of the stimulus presentation in the fMRI scanner. We have also added a depiction of the acquisition window for Experiment 1 in Figure 1D. We reproduce these below:

Figure S2. Schematic of stimulus presentation and experimental design.

Fig. S2. Schematic of stimulus presentation and experimental design.

We used a “sparse scanning” paradigm, with MR acquisitions interleaved with stimulus presentation such that the noise produced by acquisitions did not overlap with stimulus presentation. Acquisitions occurred every 2.4 seconds and in Experiments 1, 2, and 3 lasted 560 milliseconds. In Experiment 4, the acquisition lasted 870 milliseconds, as the number of slices acquired was increased to encompass the entirety of the occipital lobe to measure responses in visual cortex. The TR was therefore 2.96 seconds in Experiments 1, 2, and 3, and 3.27 seconds in Experiment 4. Stimuli were presented in a “mini-block” design, wherein the same stimulus was presented three times in a row. In pilot experiments, this design was found to yield more reliable BOLD responses given a fixed amount of scan time than an event-related design (mini-blocks of 1 stimulus presentation) or a design with five presentations per mini-block. “Foreground” sounds were 2 seconds and background noises (both real-world and synthetic) were 2.4 seconds. For mixtures of sounds and noise, the onset of the foreground sound was offset from the onset of the noise, so as to diminish the chance of perceptual grouping.

Methods: I may have missed it but did participants in Experiments 1-3 close their eyes, or look at the screen (or something else)?

They were instructed to keep their eyes open. We have added the following to the methods section to clarify: “Participants were instructed to keep their eyes open during the auditory task in all experiments (Experiment 1, 2, 3, and 4).” (Page 22, Lines 609-610)

line 591 ‘... analyses were done in this surface space, but for ease of discussion we refer to vertices as “voxels” throughout’: this is actually more confusing since values from multiple (actual) voxels will contribute to a single vertex.

The way that we mapped voxels to the cortical sheet actually upsamples, rather than downsamples the data, as there are many more vertices than voxels. By default, Freesurfer vertices are spaced about 1mm apart from one another and the functional voxels in this study are 2x2x3.08mm. Therefore, most vertices reflect the value of a single voxel. In pilot experiments we performed analyses on voxels rather than vertices and obtained similar results, so we believe the results do not depend on the choice of anatomical representation. Moreover, our sense is that “voxel” is often used in place of “vertex” in the fMRI literature (e.g. Lescroart & Gallant, Neuron 2019; de Heer, et al., J Neuro 2017; Isik, Koldewyn, Beeler, & Kanwisher PNAS 2017; Huth et al., Neuron 2012). Given all of these considerations, we respectfully prefer to maintain our terminology as it is.

Reviewer #3 (Remarks to the Author):

The study by Kell & McDermott investigates how the auditory cortex transforms a representation of the actual sound stimulus to an invariant representation of the “foreground” sound. It is suggested that primary auditory cortex is largely invariant to simple artificial “background” sounds, while only nonprimary auditory cortex is invariant to real-world “background” sounds. The results demonstrate a hierarchy of processing stages towards a invariant, object-based auditory representation. The results are solid and the manuscript is also pleasant to read. I have a few concerns, however, about the interpretation of the data.

Thank you.

1. The authors showed that the neural representations of foreground sounds are invariant to background sounds. The foreground and background sounds are defined based on acoustic stationarity, i.e., whether the statistical property changes over time. This definition, however, is somewhat arbitrary. It is certainly true that background sounds are usually stationary while foreground sounds are more dynamic, but I’m not sure if stationarity is the defining feature of foreground/background sounds. Maybe this definition is supported by the current results but it seems a bit arbitrary to use it as a priori. Furthermore, people can choose to listen to a stationary sound, e.g., rain sound or wind sound.

We agree that what is considered a “background” sound can be influenced by context and attention, as we try to emphasize in the introduction. However, it is also the case that some sounds are more informative than others on basic statistical grounds, and thus more likely to be important to the listener. There is thus longstanding interest in the possibility that these less informative (stationary) sounds are by default separated or removed. This idea is evident in prior experiments examining the robustness of cortical responses in the presence of temporally stationary noise (typically variants of white noise). We sought to extend the logic implicit in this research tradition to real-world sounds, introducing a measure of stationarity to allow us to characterize the extent to which sounds are noise-like in a statistical sense. We have sought to expand and clarify the introduction to better position the paper in the context of previous work, pointing out that we are explicitly not studying the role of goal-directed attention in selecting particular sources based on context, which we agree can be considered another means to attain noise-robustness:

“What is considered “noise” can depend on context, and thus the ability to hear sound sources of interest is in some cases critically dependent on selective attention²⁶⁻³⁰. However, some sounds are more informative than others on basic statistical grounds. Stationary signals, for instance, have stable statistics over time and thus convey little new information about the world, such that it might be adaptive to attenuate their representation relative to non-stationary sounds. There has been longstanding interest within engineering in developing methods to separate or remove these less informative (stationary) sounds from audio signals^{31, 32}. Neuroscientists have explored the possibility that the brain might by default do something similar with experiments measuring how cortical responses are affected by simple synthetic noise^{14, 17, 18, 33-35}. We sought to extend the logic implicit in this research tradition to real-world sounds^{36, 37}, introducing a measure of stationarity to characterize the extent to which natural sounds are noise-like in this statistical sense. We hypothesized that robustness to the more structured sources of noise that can be found in everyday auditory environments might necessitate mechanisms situated later in the putative cortical hierarchy.” (Page 2, Lines 45-56)

What the current results show is that, when stationary and nonstationary sounds are simultaneously presented, nonprimary auditory cortex selectively responds to the nonstationary component. Based on this interesting result, I think a more appropriate conclusion is that nonprimary auditory cortex selectively represents nonstationary sounds. Related to this point, I wonder if the authors have data about how nonprimary auditory cortex responds to independently presented nonstationary sounds. I wonder if the lack of responses to nonstationary sounds is due to feature tuning properties in nonprimary auditory cortex, i.e., the lack of neurons sensitive to stationary sounds, or due to a competition between sound sources.

We largely concur with the reviewer that this is an appealing potential explanation for the type of noise invariance that we are measuring. We appreciate the suggestion to analyze responses to isolated stationary and non-stationary sounds in primary and non-primary areas. Fortuitously, these data were collected for some of the participants in two of our experiments, and we used them to perform this analysis. As shown below, the primary

and non-primary ROI responses vary in the predicted direction: both ROIs respond more to the foreground (non-stationary) sounds than to the background (stationary) sounds, but the ratio is higher in the non-primary ROI.

Figure S9. Mean responses to foreground sounds and background noise in isolation.

Fig. S8. Mean responses to foreground and background sounds in isolation.

- (A) Mean responses to foreground sounds and real-world background noise in primary and non-primary regions from Experiment 1. Only seven of eleven subjects in Experiment 1 were presented with the background noise sounds in isolation (i.e., not superimposed on the foreground sounds). Error bars plot within-subject SEMs.
- (B) Same as (A) but for Experiment 2.
- (C) Ratio of responses the means from (A). Error bars reflect within-subject SEMs.
- (D) Ratio of means from (B).

These results are obviously consistent with the general idea that noise invariance could be driven by mechanisms that filter out stationary sounds. We think resolving this definitively will require a quantitative model. The data provide motivation for pursuing this as an exciting next step in this research program. We now note this possibility in the discussion, and reference the above analysis, which has been included as a supplementary figure.

The text now reads:

“...could noise invariance be explained by tuning to different rates and scales of spectrotemporal modulation? Because modulation tuning is prominent in primary auditory cortex⁴⁹⁻⁵², and because stationary and non-stationary sounds have distinct modulation spectra (Supplementary Fig. S1), invariance to noise could result from such noise falling “out of band”³⁴ with respect to this tuning. Consistent with this general idea, mean responses were overall higher to foreground sounds than background sounds, and the ratio between responses was higher in non-primary auditory cortex (Supplementary Fig. S9). Modeling could clarify whether the difference in invariance for synthetic and more realistic noise sources could be explained in this way.” (Page 10, Lines 251-257)

2. The authors showed a nice hierarchy of processing stages for stationary sounds, simple stationary sounds, and realistic stationary sounds. I wonder if this hierarchy is related to the hierarchy of temporal windows in different auditory processing stages. We know that low-level auditory processing stages can phase lock to very fast temporal modulations while the phase-locking ability reduces along the ascending pathway. My guess is that the synthetic stationary sounds are stationary (based on the working definition in this paper) across very short time windows (e.g., 10 ms), while the real-life stationary sounds are only stationary for longer time windows. If this is true, the selectivity to nonstationary sound may be attributed to the longer temporal integration window in nonprimary auditory cortex. Related to this point, it will be useful to show the modulation spectrum of different sets of stimuli in the supplementary figures.

We agree that selectivity for longer time scales could potentially contribute to the noise invariance that we observe. As noted in the previous response, we think definitively establishing this idea as plausible will require a significant modeling effort, and so we think the appropriate thing to do for this paper is to discuss this as an interesting possibility, which we have done in the discussion (described above). We agree that the modulation spectra of the stimuli are useful to think about in this context, and so have added the (temporal) modulation spectra for sounds in Supplementary Figure S1. These show that the stimulus classes on average have distinct modulation spectra (with the real-world noises being more lowpass than the synthetic noise, but less so than the foreground sounds), raising the possibility that filtering in this domain could potentially help to produce noise-robust representations. We reproduce them below:

Figure S1. Mean modulation spectra of stimuli.

A Logarithmically spaced modulation filters

B Linearly spaced modulation filters

Fig. S1. Modulation power in stimuli.

(A) Each panel shows the mean modulation power in each of a set of bandpass modulation filters for a set of stimuli (the right-most column, white noise, was not used as a stimulus in the study but is included for comparison). Audio waveforms were passed through a bank of bandpass “cochlear” filters, and the Hilbert envelope of each filter was passed through a second bank of (modulation) filters. These modulation filters were logarithmically spaced and tiled the modulation domain from 1-200 Hz. Each bin indicates the average power for a modulation rate within a cochlear channel.

Power is expressed in decibels (dB) and the color scale is identical across all four panels and is normalized such that the max across all four panels is set to zero.

(B) Same as (A) but with linearly spaced modulation filters that tiled the same range (1-200 Hz).

In addition to these mean differences in modulation spectra, the real-world noises exhibit variation from sound to sound in higher-order statistics. To illustrate this we have added panels to Figure 3 showing the cross-channel correlations and modulation spectra for three example real-world noises and their synthetic counterparts (compare two rows of panel B):

(A) Example cochleograms of real-world background noises and spectrally matched synthetic noises. Right: Average spectrum.

(B) Statistics of example real-world noises (top row) and spectrally matched synthetic noises (bottom row). Subpanels show cross-channel correlations (left) and modulation spectrum (right).

To summarize these differences, we have added a supplementary figure showing the distances between the two types of noise stimuli and white noise, computed using either the power spectrum, cross-channel correlations, and modulation spectra. This analysis shows that both types of noise differ spectrally for white noise, but that the synthetic noise is more similar to white noise than to the real-world noise in both types of higher-order statistics.

Figure S5. Statistical differences between types of noise.

Average distance in statistics between real-world noise and the corresponding spectrally-matched synthetic noise. For reference, we have included distances for each of these noises with white noise, which was not included as a stimulus, but was included in this figure to offer a structureless reference. As expected, both real-world and spectrally-matched synthetic noise have similar frequency spectra (left), both of which are dissimilar from white noise. By contrast, the spectrally matched synthetic has cross-channel correlations (middle) and modulation spectra (right) that are much more similar to white noise than to real-world noises. Error bars plots SEMs across sounds.

Furthermore, in response to the reviewer’s suggestion about stationarity in finer time bins, we have measured the stationarity of the real-world “foregrounds”, the real-world “noise”, and the synthetic noise, using a range of averaging window going down to 10 milliseconds. As shown below, there are differences between the two classes of noise stimuli in the expected direction (slightly more variability in statistics across windows for the real-world noise), but they are small. The real-world noises are quite stationary because they were selected in this basis; both types of noise stimuli are close to the stationarity of white noise. The differences in stationarity thus seem unlikely on their own to explain the neural effect. We think the different effects of synthetic and real-world noise are probably due to the fact that the statistics of real-world noise are less noise-like (as shown in the new Fig. 3B, and the new Supplementary Fig. 5), such that the real-world noise may be more confusable with non-stationary sounds on a local time scale. This could translate to needing to integrate over a longer timescale in order to determine what features in a sound mixture are due to a stationary signal.

Because the differences between noise stimuli are evident in the statistic analyses shown above, we have opted to include those in the manuscript, but provide the stationarity analysis below:

Stationarity score by temporal window length

Stationarity score as a function of window length for averaging.

We have also added a line to the discussion on this topic:

“Given that modulation tuning accounts for little of the neural responses outside primary auditory cortex in humans⁵³, and given that modulation spectra of real-world sounds vary considerably from sound to sound³⁷ (Fig. 3B), it seems unlikely that the increase in noise invariance between primary and non-primary cortex could be entirely explained with modulation filtering.” (Page 10-11, Lines 258-261)

Minor points:

1. Please show the boundary of the scanned area and the boundary of nonprimary auditory cortex in the figures.

We have added panel 1D that shows this, and reproduced it below:

2. *Experiment 2 is done to exclude speech and music. I wonder why this is necessary. They could be excluded based on the data from Experiment 1. Is it because Experiment 1 has too many speech/music sounds?*

Yes, exactly. We have updated the text to help clarify this issue:

“Experiment 1 left this question open because about half of the foreground sounds were instances of speech or music. Given the relatively small number of non-speech and non-music sounds in the data from Experiment 1, we instead ran a second experiment.” (Page 5, Lines 115-117).

3. *Line 170, how about nonprimary auditory cortex? Are the responses changed by the task?*

We have examined the mean responses in primary and non-primary auditory cortex as they are affected by the visual and auditory task. We have added the following language to the main text we have included the results as Supplementary Figure S8, reproduce below:

4. *For the short time windows used in the current study, it's not meaningful to discuss very low-frequency modulations, e.g., 0.5 Hz modulations (Line 438).*

We agree with the reviewer, and we only included modulation power from filters whose center frequency completed a cycle in each segment and thus always excluded many of the low-frequency modulations. We explained that later in the methods section, but to preempt any confusion, we have updated the text to note this at the point where we introduce the modulation filters:

“For the stationarity measure described below, we only included power from modulation filters whose center frequency completed a cycle in each segment (e.g., for the 100-millisecond segments we excluded all filters with center frequencies less than 10 Hz).” (Page 19, Lines 488-490)

Reviewers' Comments:

Reviewer #1:

Remarks to the Author:

The authors have done a very good job of responding to the comments, including providing additional data, some of which will be included as supplementary material. I am quite satisfied and think the paper makes an excellent contribution.

I have only one remaining comment regarding the attentional manipulation. My critique before was that the cross-modal manipulation used did not really address the issue of whether some of the results could be explained by differences between the "attention-grabbing" aspect of the foreground vs background noise. The authors seem to agree, and provided extensive information about other approaches they have considered to address this question. I certainly did not expect them to run a new study to address the issue. But I do think that perhaps the relevant section of the discussion could be modified to take this point into account.

The section in question reads "Third, we show that noise robustness is present even when attention is directed elsewhere. Non-primary cortical responses have been observed to be modulated by attention to one speech stream over another^{16, 27, 239}, illustrating one way in which cortical responses can achieve invariance to behaviorally irrelevant sounds. It has remained unclear whether robustness to temporally stationary noise might be explained in this way, particularly given that non-stationary sounds might preferentially draw selective attention⁴²⁻⁴⁴. The effects we report here are relatively unaffected by inattention and thus suggest a form of noise robustness largely distinct from the attentional selection of task-relevant signals. "

I think this section could be modified to acknowledge that the cross-modal manipulation still leaves open the possibility that nonstationary sounds may draw more attention. They do show that the effects are robust to attention in the sense that when attention is directed to another modality the effects remain. That is important, but does not completely address the question. So I think a small mention of this point is warranted.

Reviewer #2:

Remarks to the Author:

Thanks to the authors for the very thorough revision and response, they addressed all of my points, and I thought those of the other reviewers.

Very minor detail: regarding the vertex/voxel question, from the response, it sounds like nearest-neighbor sampling was used in mri_vol2surf (or whatever script called it) rather than trilinear interpolation? And what cortical sampling scheme was used? (e.g., average across some fraction of cortical thickness?)

Reviewer #3:

Remarks to the Author:

The authors had successfully addressed my previous concerns.

In the newly added Fig. S8, lines in panel A and panel B are not labeled. Also, the figure title says Figure S9.

Response to the Reviewers

Reviewer #1 (Remarks to the Author):

The authors have done a very good job of responding to the comments, including providing additional data, some of which will be included as supplementary material. I am quite satisfied and think the paper makes an excellent contribution.

Thank you for taking the time to review our manuscript again.

I have only one remaining comment regarding the attentional manipulation. My critique before was that the cross-modal manipulation used did not really address the issue of whether some of the results could be explained by differences between the "attention-grabbing" aspect of the foreground vs background noise. The authors seem to agree, and provided extensive information about other approaches they have considered to address this question. I certainly did not expect them to run a new study to address the issue. But I do think that perhaps the relevant section of the discussion could be modified to take this point into account.

The section in question reads,

"Third, we show that noise robustness is present even when attention is directed elsewhere. Non-primary cortical responses have been observed to be modulated by attention to one speech stream over another^{16, 27 29}, illustrating one way in which cortical responses can achieve invariance to behaviorally irrelevant sounds. It has remained unclear whether robustness to temporally stationary noise might be explained in this way, particularly given that non-stationary sounds might preferentially draw selective attention⁴²⁻⁴⁴. The effects we report here are relatively unaffected by inattention and thus suggest a form of noise robustness largely distinct from the attentional selection of task-relevant signals."

I think this section could be modified to acknowledge that the cross-modal manipulation still leaves open the possibility that nonstationary sounds may draw more attention. They do show that the effects are robust to attention in the sense that when attention is directed to another modality the effects remain. That is important, but does not completely address the question. So I think a small mention of this point is warranted.

We agree with this point and have updated the manuscript. The new portion is in bold:

"Third, we show that noise robustness is present even when attention is directed elsewhere. Non-primary cortical responses have been observed to be modulated by attention to one speech stream over another^{16, 27 29}, illustrating one way in which cortical responses can achieve invariance to behaviorally irrelevant sounds. It has remained unclear whether robustness to temporally stationary noise might be explained in this way, particularly given that non-stationary sounds might preferentially draw selective attention⁴²⁻⁴⁴. The effects we report here are relatively unaffected by inattention and thus suggest a form of noise robustness largely distinct from the attentional selection of task-relevant signals. The invariance observed here may be more akin to previously reported aspects of sound segregation that are also robust to inattention^{47,48}. **However, because we manipulated attention across modalities, rather than within audition, we cannot exclude the possibility that residual attentional resources are captured more by non-stationary sounds, and that this contributes to the effects we observed.**"

We also added a sentence to the results section to clarify the choice we made regarding our attentional manipulation:

“We chose to manipulate attention across modalities because we had difficulty devising a task that could verifiably direct attention within the auditory modality to the foreground sound or background noise.”

Reviewer #2 (Remarks to the Author):

Thanks to the authors for the very thorough revision and response, they addressed all of my points, and I thought those of the other reviewers.

Thank you for taking the time to review our manuscript again.

Very minor detail: regarding the vertex/voxel question, from the response, it sounds like nearest-neighbor sampling was used in mri_vol2surf (or whatever script called it) rather than trilinear interpolation? And what cortical sampling scheme was used? (e.g., average across some fraction of cortical thickness?)

We averaged across the cortical ribbon (the normal to the surface at a given vertex). The average was computed over the values at six points: the pial boundary, the white matter boundary, and four evenly spaced locations between the two. We used trilinear interpolation, but given that moving to voxels to vertices in our case was upsampling more often than not a single voxel contributed essentially all of the signal at a given vertex.

We have added a sentence to the methods to clarify this issue:

“The value for each point on the surface was computed as the average of the (linearly interpolated) value at six points across the cortical ribbon: the pial boundary, the white matter boundary, and four evenly spaced locations between the two.”

Reviewer #3 (Remarks to the Author):

The authors had successfully addressed my previous concerns.

Thank you for taking the time to review our manuscript again.

In the newly added Fig. S8, lines in panel A and panel B are not labeled. Also, the figure title says Figure S9.

It seems we mislabeled the figure in the rebuttal (but not the Supplementary Information) – the figure is indeed Supplementary Figure 9. However, it did not have the lines labeled, and we have fixed that.